# Reward expectation extinction restructures and degrades CA1 spatial maps through loss of a dopaminergic reward proximity signal

Seetha Krishnan [1,2], Chad Heer [1,2], Chery Cherian[1] & Mark E. J. Sheffield [1] ✉

Hippocampal place cells support reward-related spatial memories by forming a cognitive map that over-represents reward locations. The strength of these memories is modulated by the extent of reward expectation during encoding. However, the circuit mechanisms underlying this modulation are unclear. Here we find that when reward expectation is extinguished in mice, they remain engaged with their environment, yet place cell over-representation of rewards vanishes, place field remapping throughout the environment increases, and place field trial-to-trial reliability decreases. Interestingly, Ventral Tegmental Area (VTA) dopaminergic axons in CA1 exhibit a ramping reward-proximity signal that depends on reward expectation and inhibiting VTA dopaminergic neurons largely replicates the effects of extinguishing reward expectation. We conclude that changing reward expectation restructures CA1 cognitive maps and determines map reliability by modulating the dopaminergic VTA-CA1 reward-proximity signal. Thus, internal states of high reward expectation enhance encoding of spatial memories by reinforcing hippocampal cognitive maps associated with reward.

Individual pyramidal cells in the hippocampus fire action potentials in specific regions of an environment, known as their place field[1]. Specific populations of place cells represent cognitive maps of specific environments[2,3]. As animals become familiar with an environment, hippocampal place fields become more stable and many of them are reinstated whenever the animal navigates the environment[4–6]. When environmental cues change, hippocampal place cells "remap" through changes in firing rate and place field locations, thought to be an integral part of navigational and memory processes[2,3,5,7]. Place cells and the maps they comprise allow animals to acquire, store, code, and recall environments[8,9]. In addition, place cells are modulated by external context variables that include odors and colors[10] but also include internal context variables such as attention[11,12], decisions of future trajectories[13–15] and fear[16,17]. Reward expectation (or reward prediction) is another internal context variable that could modulate cognitive maps as it has a powerful influence on hippocampal-dependent memories[18–20]. However, the influence of reward expectation on

cognitive maps remains unclear and some evidence suggests cognitive maps may be independent from reward expectation[21–23].

Rewards themselves are represented through a distinct population of reward cells[24] and an over-representation of place cells tuned to rewarded locations in CA1[25–29]. The over-representation of reward locations by place cells requires learning[27] and is dependent on the probability a reward will be delivered at those locations[28]. This suggests that it is reward expectation that determines over-representation of reward locations rather than the attainment of the reward itself, although this remains to be determined. Furthermore, cues distant from reward locations can predict the attainment of future reward, i.e., lead to them[30]. Therefore, over-representation of reward locations by place cells does not explain how reward expectation might influence the encoding of locations that are distant from, but lead to, rewards. What is also unclear is how changes in reward expectation within an environment might influence the place cell code in the hippocampus, or the time course over which such changes may occur.

[1]Department of Neurobiology and Institute for Neuroscience, University of Chicago, Chicago, IL 60637, USA. [2]These authors contributed equally: Seetha Krishnan, Chad Heer. ✉e-mail: sheffield@uchicago.edu

The learned associations between isolated cues and future rewards involves dopaminergic circuits that respond to reward predicting cues and act as a learning signal[31]. Dopaminergic circuits that encode reward expectation project to the hippocampus from the ventral tegmental area (VTA)[32] and influence hippocampal function, synaptic plasticity, and synaptic transmission[29,33,34]. Recent work has begun to elucidate the role of reward expectation and dopamine circuits during spatial navigation and has uncovered a spatial proximity to reward signal emanating from the VTA that ramps up in activity as animals approach an expected reward location[30,35–38]. Like the classic studies of reward learning, spatial locations during navigation that predict future reward trigger increases in dopamine release. As the animal approaches the reward, dopamine levels ramp up as locations closer to reward are better predictors of the reward, i.e., they increase reward expectation. This suggests that reward expectation may influence the encoding of space during navigation through the effects of dopamine modulation, possibly through ramping dopamine signals[37], but whether these signals exist in the hippocampus has yet to be demonstrated experimentally, as is their influence on cognitive maps.

Spatial memories encoded in the hippocampus are represented by populations of place cells that create a cognitive map, and reward expectation influences these memories through dopaminergic circuits[18–20,23,33,34]. We therefore hypothesized that changing reward expectation would modulate place cell properties and transform the structure of the cognitive map (remapping) driven by dopaminergic signals from the VTA. However, testing this hypothesis is challenging as manipulating rewards to change reward expectation alters navigation behaviors that affects place cells[39]. We therefore developed a paradigm that changes reward expectation in head-restrained mice repeatedly traversing an unchanging virtual linear track. Importantly, mice in this setup showed matched navigation behaviors and engagement with their environment across many trials during changes in reward expectation, allowing us to isolate the influence of reward expectation on place cells. This head-restrained setup allowed for continuous 2-photon calcium imaging of large populations of place cells in CA1 as well as direct calcium imaging of dopaminergic axons from VTA in CA1 during changes in reward expectation. We also inhibited dopaminergic neurons in the VTA to further test our hypothesis.

## Results

### Lowering reward expectation changes spatial encoding of an unchanging spatial environment in CA1

Mice were trained to run on a treadmill along a 2 m virtual linear track for water rewards (rewarded condition: R) delivered at the track end (Fig. 1a, b), after which they were teleported back to the start of the track. Well-trained mice learned the location of the reward and preemptively licked before the reward location (pre-licking), providing a lap-wise behavioral signal of reward expectation (Supplementary Fig. 1). On experimental day (Fig. 1b), mice ran in R for 10 min before water reward was unexpectedly removed (unrewarded condition: UR). Interestingly, mice continued pre-licking for a few laps in UR, as though still expecting a reward (see "Licking behavior" section in "Methods"; Fig. 1ciii and Supplementary Fig. 1). After mice traversed UR for 10 min, reward was reintroduced (re-rewarded condition: RR).

Additionally, using 2-photon calcium imaging of dorsal CA1 pyramidal neurons expressing the genetically encoded calcium indicator GCaMP6f[40] (Fig. 1a), we measured population activity while mice were switched across conditions: R-UR-RR (Fig. 1b, c). Behavior and activity in an example mouse are shown in Fig. 1c. We found that removing reward caused a dramatic change in population activity (Fig. 1ci, population activity is represented as a raster plot where cells with correlated activity are arranged next to each other, see Methods). This was not a consequence of time or running behavior (Supplementary Figs. 2 and 3) and like changes in pre-licking, did not occur immediately

after reward removal (Fig. 1c). To quantify this, we trained a naive Bayesian classifier with all the extracted cells on the initial laps in R and used the trained classifier to predict track position on the final laps of R and all laps in UR and RR (Methods). We found that the decoder was able to accurately predict position on the final laps in R and initial laps in UR before abruptly underperforming (Fig. 1d). Because mice pre-licked for a few laps in UR, we asked if decoder underperformance was associated with reduced reward expectation. In all mice ($n = 12$), we quantified the average pre-licking and decoder fit on each lap after reward removal by running a rolling average (Fig. 1e, see "Methods"). On average, pre-licking continued for a few laps before rapidly dropping, and interestingly, decoder performance sharply dropped around the same lap when pre-licking reached zero. This decreased decoder accuracy with decreased pre-licking indicates that hippocampal spatial encoding remains unchanged following reward removal, until reward expectation diminishes, at which point the spatial code abruptly transforms.

To further quantify this, we identified the lap on which pre-licking stopped in each mouse (Methods). For clarity, we labeled the laps with pre-licking as having high reward expectation ($RE_{high}$) and the laps after pre-licking stopped as having low reward expectation ($RE_{low}$). Indeed, decoder accuracy in $RE_{high}$ laps was similar to R and was significantly lower in $RE_{low}$ laps (Fig. 1f, mean decoder $R^2$ [95% confidence intervals (CIs)]: R = 0.95 [0.93 0.97], $RE_{high}$ = 0.90 [0.87 0.93], $RE_{low}$ = 0.65 [0.54 0.75]). This held true independent of our definition of when licking stopped (Supplementary Fig. 4). The decoder accuracy somewhat recovered following reward re-introduction in RR laps, although it remained lower than in R (RR = 0.82 [0.77 0.87]). Reduced decoder performance during $RE_{low}$ was not explained by differences in time or running velocity (Supplementary Figs. 2 and 5).

To quantify these changes further, we analyzed decoder error across the track by measuring the absolute distance between the true position from the position predicted by the decoder at each point on the track. Interestingly, in $RE_{low}$, decoder error had increased at all locations across the track (Fig. 1g), and not just around the reward site, as may have been predicted[25–29]. As observed before, in RR, decoder error decreased across the track but remained lower than in R (Fig. 1g). These data provide evidence against spatial encoding being independent from reward expectation[21] and demonstrate that changing reward expectation drastically alters spatial encoding at all locations within an unchanging spatial environment.

### Changes in spatial encoding associated with diminished reward expectation are not due to disengagement with the environment

Next, we tested an alternate explanation; that disengagement with the environment in $RE_{low}$ laps was responsible for the changes in spatial encoding[11,12,41]. We noticed that mice in R slowed down as they approached the reward site, exhibiting engagement with the VR environment[24] (Supplementary Figs. 6 and 7). To confirm this, we exposed mice to a dark environment without any virtual cues and indeed found an absence of this approach behavior ($n = 6$, Supplementary Fig. 6A). We therefore interpret approach behavior on each lap as a behavioral readout of engagement with the environment. In UR we observed changes in velocity throughout the track, but approach behavior remained intact on most laps in UR ($n = 12$, 170/244 laps, 70%). This was true even after mice stopped licking ($RE_{low}$: Engaged; Supplementary Figs. 6 and 7). In contrast, laps displaying disengagement with VR were less frequent ($RE_{low}$: Disengaged, 74/244, 30%, Supplementary Figs. 6 and 7). Importantly, we found a similar reduction in decoder performance in UR when using only $RE_{low}$ engaged laps (Supplementary Fig. 6B, Mean decoder $R^2$ [95% CI]: R = 0.95 [0.93 0.97], $RE_{high}$ = 0.90 [0.87 0.93], $RE_{low}$: Engaged = 0.65 [0.49 0.81]), although disengaged laps did further reduce decoder performance ($RE_{low}$: Disengaged = 0.31 [0.06 0.5]). The disengaged laps were

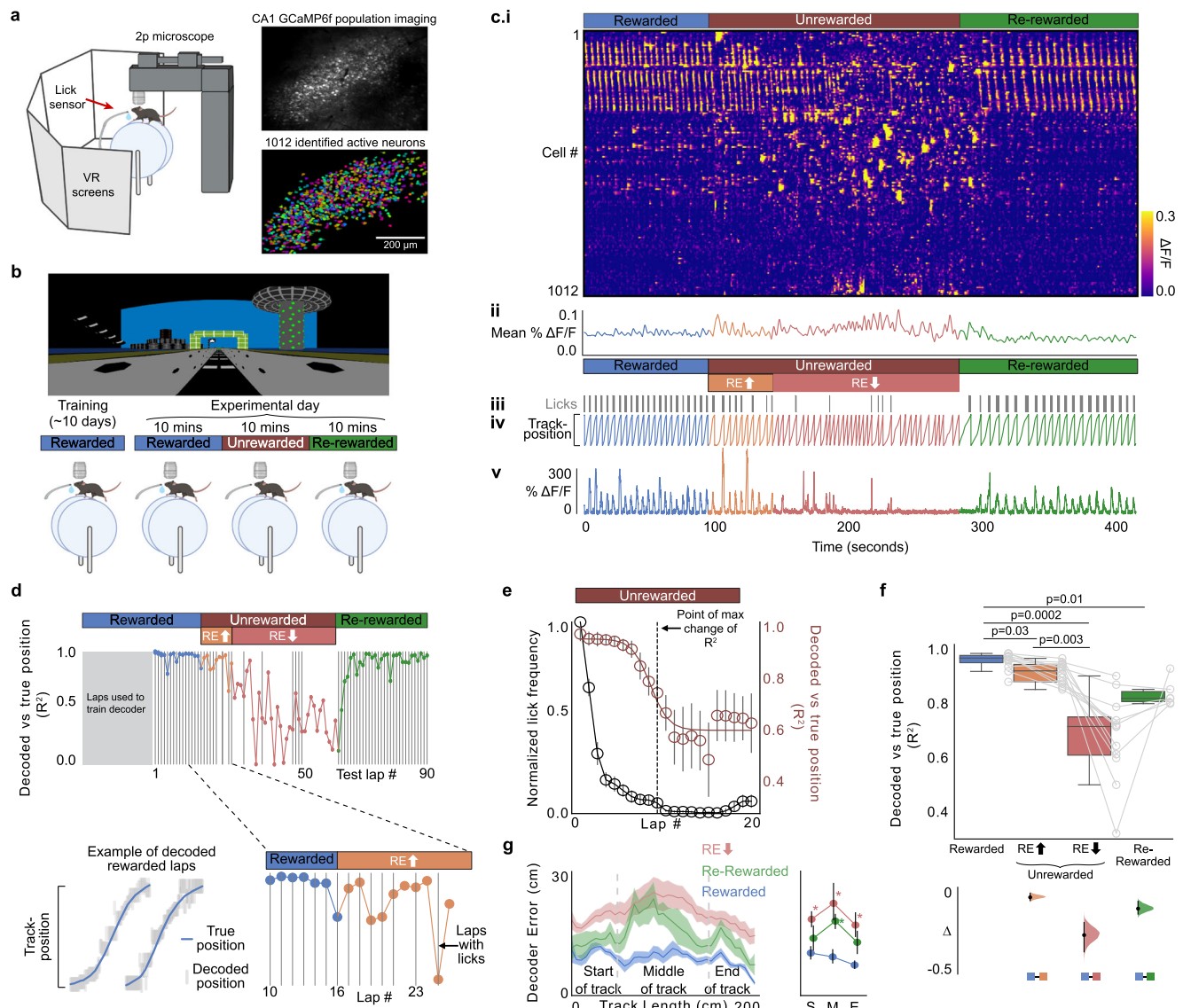

**Fig. 1 | Diminished reward expectation restructures spatial encoding in the hippocampus. a** Experimental setup (left), created with BioRender.com. A typical field of view in CA1 (right, top). Extracted regions of interest, randomly colored (right, bottom). **b** Experimental protocol. Image of virtual track (top). Changing reward contingencies (bottom, Rewarded (R), Unrewarded (UR), Re-Rewarded (RR)), created with BioRender.com. **c** i: Rasterplot representing fluorescence changes ($\Delta F/F$) of cells in A across time. Cells (y-axis) are arranged with the most correlated cells next to each other. ii: Mean $\Delta F/F$ of the cells in (i). iii: Mouse licking behavior. iv: Mouse track position. v: $\Delta F/F$ from an example cell. Laps before animal stops consistently licking in UR were considered laps with high reward expectation ($RE_{high}$, orange laps) and after licking stops are laps with low reward expectation ($RE_{low}$, red laps, see Methods). **d** A Bayesian decoder was trained on CA1 activity from initial laps in R and tested on remaining laps. (Top) Coefficient of determination ($R^2$) between true and predicted position of tested laps, (bottom right) zoomed in. (Bottom left) An example fit. Gray lines indicate laps with licks. **a–d** are from the same animal. **e** Mean decoder $R^2$ fitted with a reverse Boltzmann Sigmoid ($r = 0.94$; magenta), mean lick frequency normalized to maximum licks (black) for each lap in unrewarded condition. Error bars indicate s.e.m. The point of maximum change in $R^2$ as calculated from the fit is indicated by the dashed line. **f** (Top) Boxplots (see methods for definition) show distribution of mean decoder $R^2$ in the different conditions. Circles represent individual animals. $P$ values were obtained using a two-sided Paired $t$ test. (Bottom) Bootstrapped mean differences ($\Delta$) with 95% Confidence Intervals (CI) (error bar). $X$-axis indicates the comparisons made. **g** (Left) Mean decoder error by track position. Shading indicates s.e.m. (Right) Mean decoder error binned by track position as indicated by gray bars in the left panel. Error bars indicate 95%CI. S: Start of the track, M: Middle of the track, E: End of the track. Asterisk (*) indicates significant $p$ values ($P < 0.01$, two-sided paired $t$ test) obtained by comparing $R$ with other conditions at each position. $n = 12$ mice used for **f**, **g**.

uniformly distributed throughout $RE_{low}$ (see example laps in Supplementary Fig. 7). The proportion of disengaged laps also did not progressively increase with time, as was observed in Petit et al.[41]. The probability of disengaged laps in the first half of $RE_{low}$ was similar to the second half ($P = 0.14$). Furthermore, the distribution of disengaged laps throughout the session did not differ from a uniform distribution ($P = 0.7$). The influence of disengagement on CA1 spatial representations is in agreement with a recent paper[41], but is not the main focus of this paper.

Using an alternate method to measure engagement with the VR environment, we quantified pupil area during running in R and UR conditions as pupil diameter has been shown to be a measure of attentional/arousal state ($n = 5$, Supplementary Fig. 6C–F)[42,43]. We observed a distinct pattern of pupil area changes during laps in R which included an increase in pupil area near the end of the track. To quantify if within lap pupil area dynamics were altered in UR compared to R, we calculated the Pearson's correlation coefficient of each lap's pupil area dynamics to the mean pupil area dynamics from all laps in R.

We first ensured that any changes in pupil area correlation in UR were not due to changes in animal behavior in UR (Supplementary Fig. 8). Engaged laps in UR in each animal were then defined as laps where the correlation coefficient was greater than or equal to the mean correlation coefficient of laps in R (the remaining laps were defined as disengaged laps). 71% of these engaged laps were also classified as engaged laps using the approach behavior described above (57/80 laps; chance levels = 37%). We again found that animals were engaged on most laps in UR ($n = 5$, 80/106, 75%) after licking stopped ($RE_{low}$: Engaged, Supplementary Fig. 6E) and found a similar reduction in decoder performance when using only these laps (Supplementary Fig. 6F, mean decoder $R^2$ [95% CI]: $R = 0.91$ [0.81 1.00], $RE_{high} = 0.83$ [0.71 0.94], $RE_{low}$: Engaged = 0.57 [0.34 0.79], $RE_{low}$: Disengaged = 0.43 [0.13 0.74]). Therefore, using two distinct measures of engagement with the VR environment, approach behavior and pupil area, we conclude that changes in spatial encoding in UR are not due to disengagement with the VR environment but instead are due to lowered reward expectation. However, because disengagement further influences spatial encoding, we focused further analysis on engaged laps (laps with approach behavior) during $RE_{low}$ to isolate the effects of $RE_{low}$ without confounds introduced by disengagement with the VR environment[41].

## Lowering reward expectation induces place cell remapping

The changes in spatial encoding in UR suggests changes in activity patterns of place cells (remapping). Place cell (PC) remapping is linked to encoding changes within environments or distinct environments as well as separated exposures to the same environment across days[2,4,5,7,44]. We explored if changes in reward expectation may induce remapping in an unchanging spatial environment within a single session. We first defined PCs in R and constructed a population firing vector using these cells for each lap in R-UR-RR and correlated these vectors across the session using only those laps in UR that showed engagement with the VR (only $RE_{low}$: Engaged laps, Fig. 2a). Correlations showed pronounced transitions in population activity around the lap when licking stops ($RE_{high}$ to $RE_{low}$) and again following reward reintroduction ($RE_{low}$ to RR). To adequately quantify these transitions, we clustered the lap-by-lap PV correlations and calculated the probability of laps being part of the cluster to which R laps belong (see "Methods"). The lap-wise cluster probability traces revealed dips in probability when transitioning from $RE_{high}$ to $RE_{low}$ and not between R and $RE_{high}$ (Fig. 2a bottom; mouse 3 correlation drops a lap before our definition of low reward expectation) confirming changes in place cell activity occurring as animals stop licking and not immediately following reward removal. Moreover, the probability increased to a certain extent in RR laps. This is most apparent in mouse 1 which comes all the way back to R levels. However, mice 2 and 3 did not show a similar reinstatement nor was there an immediate transition in cluster probability following reward reintroduction.

We next analyzed the $RE_{low}$ period in UR in these 3 mice as they had sufficient numbers of engaged laps in all conditions (R-UR-RR) to define place fields using only the engaged laps and removing disengaged laps (see Supplementary Fig. 9 for analysis on all mice using both engaged and disengaged laps). We found $RE_{low}$ caused partial remapping as shown by a drop in the spatial correlation of place fields between R and UR conditions (using only $RE_{low}$: Engaged laps; Fig. 2b–d). Interestingly, partial remapping caused by $RE_{low}$ occurred at all locations throughout the environment, and not just at locations near the reward site (Fig. 2d). As observed with the population firing vector correlations, the extent of remapping was reduced in RR−i.e., the R map seemed to somewhat return in RR (Fig. 2b–d).

To further analyze place field dynamics, we determined the fate of individual place fields throughout R-UR ($RE_{low}$: Engaged laps only)-RR. We found 27.9% of place fields found in R (605 PFs in R, $n = 3$) remained stable throughout R-$RE_{low}$-RR (169/605; Fig. 2e blue throughout). In

$RE_{low}$, R place fields either remained stable (222/605, 36.7%; Fig. 2e blue, middle column), disappeared (249/605, 41.2%; Fig. 2e black, middle column), or remapped (134/605, 22.1%; Fig. 2e cyan, middle column). In addition, new place fields formed in $RE_{low}$ (296/663 of all place fields found in $RE_{low}$, 44.6%; Fig. 2e red, middle column). More R-place fields retained their fields in RR (311/605, 51.4%; Fig. 2e blue, right column), but other R-place fields remapped in RR (89/605, 14.7%; Fig. 2e cyan, right column). Furthermore, of the place fields that disappeared in $RE_{low}$ some reappeared in RR (126/249; 50.6%; Fig. 2e blue adjacent to black, right column). Only a small proportion of those that were newly formed in $RE_{low}$ remained stable in RR (64/296 new PFs in $RE_{low}$; 21.6%; Fig. 2e red, right column) and a larger percentage disappeared in RR (204/296; 68.9%; Fig. 2e black adjacent to red, right column). Finally, of the 674 place fields found in RR, 140 were newly formed (20.8%; Fig. 2e green, right column). These place field dynamics were also observed when place fields from all mice were analyzed that included engaged and disengaged laps (Supplementary Fig. 9) and were not observed in control mice that stayed in R for a matched number of laps (Supplementary Fig. 10). Example of the lap-by-lap dynamics of individual place cells throughout R-UR-RR are shown in Fig. 2f. In summary, transitioning to $RE_{low}$ restructures the CA1 place code at all locations within an unchanging spatial environment through the disappearance, emergence, and remapping of place fields. Although a component of the structure returns when transitioning back to RR, the original structure remains changed, and further restructuring takes place. This suggests that the CA1 spatial code of an environment is dependent on reward expectation and the history of reward expectation. This supports the idea that CA1 performs context discrimination in an unchanging spatial environment.

## Place fields show diminished reliability and increased out-of-field firing following lowered reward expectation

The quality of hippocampal spatial encoding is related to memory performance[11,45]. Qualitatively, increased reliability of firing across multiple traversals through the same location, low out-of-field firing, and decreased place field width are general indicators of better spatial encoding accuracy. Measuring these properties, we asked whether place fields in $RE_{low}$ were of the same or reduced quality compared to place fields in R. Plotting place fields defined in each condition, we found a total number of place fields that tiled the track to be similar in R, $RE_{low}$ (using only Engaged laps), and RR ($n = 3$, Fig. 3a, findings from all animals using both engaged and disengaged laps are shown in Supplementary Fig. 11). We quantified place cell properties and found that the place fields in $RE_{low}$ had degraded on every measure of spatial encoding we used (see Methods): place field trial-to-trial reliability[23] (Rewarded = 0.47 [0.38 0.57], $RE_{low}$: Engaged = 0.32 [0.15 0.48]), out/in place field firing ratio (Rewarded = 0.10 [0.07 0.13]; $RE_{low}$: Engaged = 0.16 [0.12 0.20]) and place field width (Rewarded = 53.34 [43.87 62.81]; $RE_{low}$: Engaged = 60.56 [48.07 73.06]), across all locations (Fig. 3b). There was also a small decrease observed in firing intensity (Rewarded = 0.36 [0.31 0.40]; $RE_{low}$: Engaged = 0.31 [0.23 0.40]). These place field properties returned to or approached R levels in RR, except firing intensity which remained low in RR (Fig. 3b, Reliability 0.41 [0.31 0.51]; Out/In Field Firing 0.12 [0.08 0.15]; Place field width (cm) 54.08 [46.86 61.31]; Firing Intensity 0.29 [0.24 0.34]). The degradation of place fields in $RE_{low}$ and return in RR was also observed when we included engaged and disengaged laps together (Supplementary Fig. 11) and was not due to time (Supplementary Fig. 12). This demonstrates that diminished reward expectation leads to a spatial code in CA1 with low quality place fields at all locations, suggesting a weakened spatial memory representation of the environment[11,45].

Finally, we compared the degradation of place cells following reward expectation extinction to a novel never-reinforced environment (Supplementary Fig. 13). We found that the spatial decoding and place cell parameters were poor in the initial laps of the novel environment,

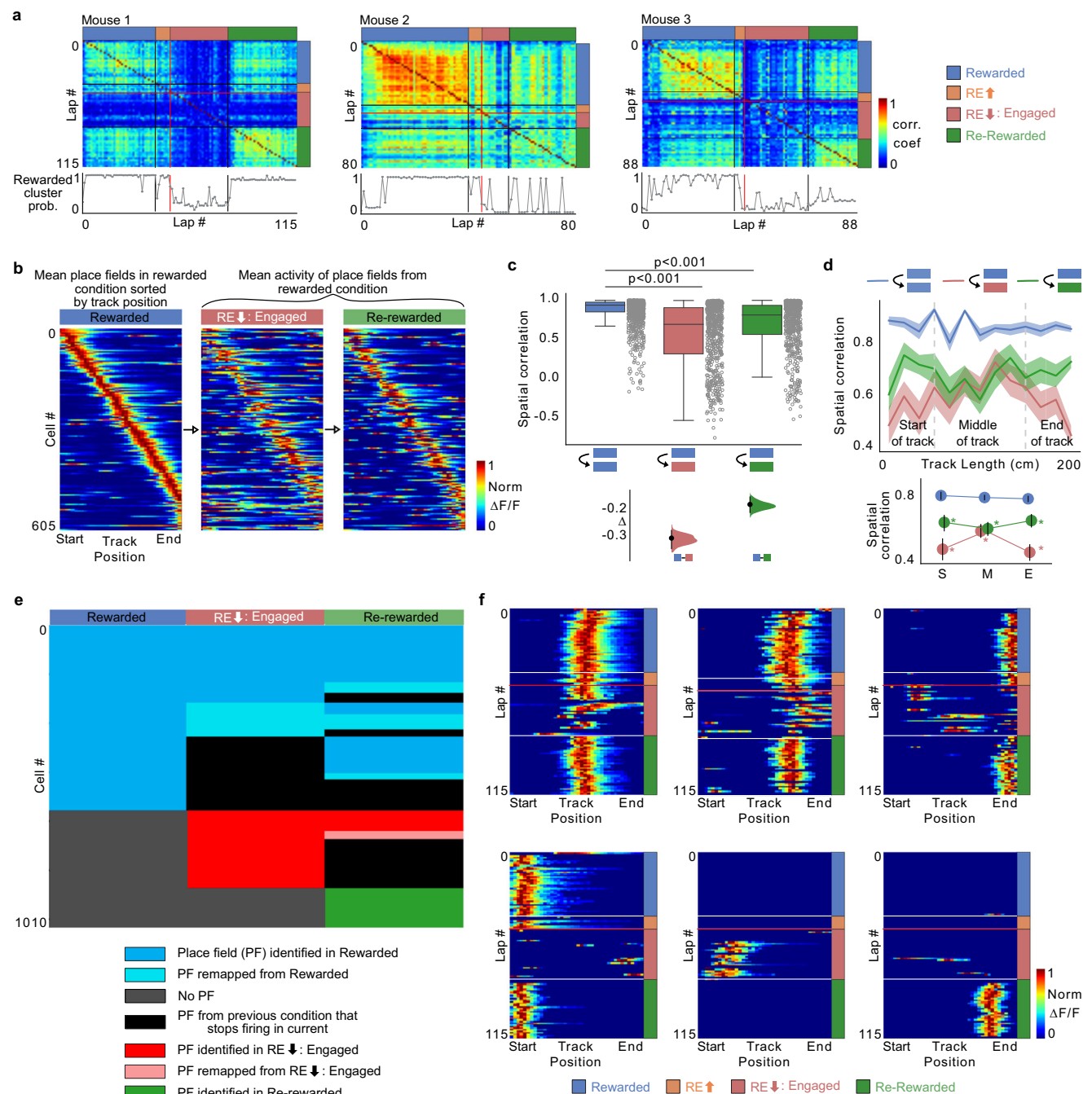

**Fig. 2 | Diminished reward expectation leads to place cell remapping across all track locations and loss of reward overrepresentation. a** (Top) Population vector correlations between place cell activity in each lap in all conditions in 3 animals. In laps with low reward expectation, only Engaged laps were used (RE$_{low}$: Engaged). (Bottom) Probability of laps belonging to the Rewarded cluster following 1000 iterations of $k$-means clustering (see Methods). Red line indicates the lap where pre-licking stops, and black lines divide each condition. **b** Place fields defined in R plotted across all conditions. Activity of each place cell was normalized to peak in R and sorted by its center of mass along the track. **c** Boxplots show distribution of place field spatial correlation for cells (circles) within R condition (blue) and between R and other conditions. *P* values were obtained using two-sided paired *t* test. (Bottom) Bootstrapped mean differences (Δ) with 95% CI (error bar). *X*-axis

indicates the comparisons made. **d** (Top) Same data, averaged by track position. Shading indicates s.e.m. (Bottom) Average correlation binned by track position indicated by gray lines in the top panel. S: Start of the track, M: Middle of the track, E: End of the track. Asterisk (*) indicates significant *P* values (two-sided paired *t* test, *P* < 0.01) obtained by comparing R (blue) with other tasks at each position. Both **d**, **e** use *n* = 605 place cells defined in R. **e** Fate of place cells identified in different conditions. Place fields identified in R (blue) can be stable throughout RE$_{low}$ (blue) and RR (blue). They can also remap in RE$_{low}$ and RR (light blue) or lose their place field completely (black). New place fields can form in RE$_{low}$ (red) and be stable (red) or remap (light red), in RR. New place fields can also form in RR (green). **f** Example place cell activity in the different conditions. White lines divide each condition and the red line indicates lap defined as when pre-licking stops.

however, with time, as the animal learned the environment and developed reward expectation (Supplementary Fig. 13B), these properties became better and comparable to properties in R and RR (Supplementary Fig. 13C–E). Both reliability and out/in field firing ratio in

RE$_{low}$ matched the levels of the initial trials in the novel environment and remained so throughout the session until reward expectation was reinstated in RR (Supplementary Fig. 13C–E). This further demonstrates that reward expectation enhances the spatial code in CA1.

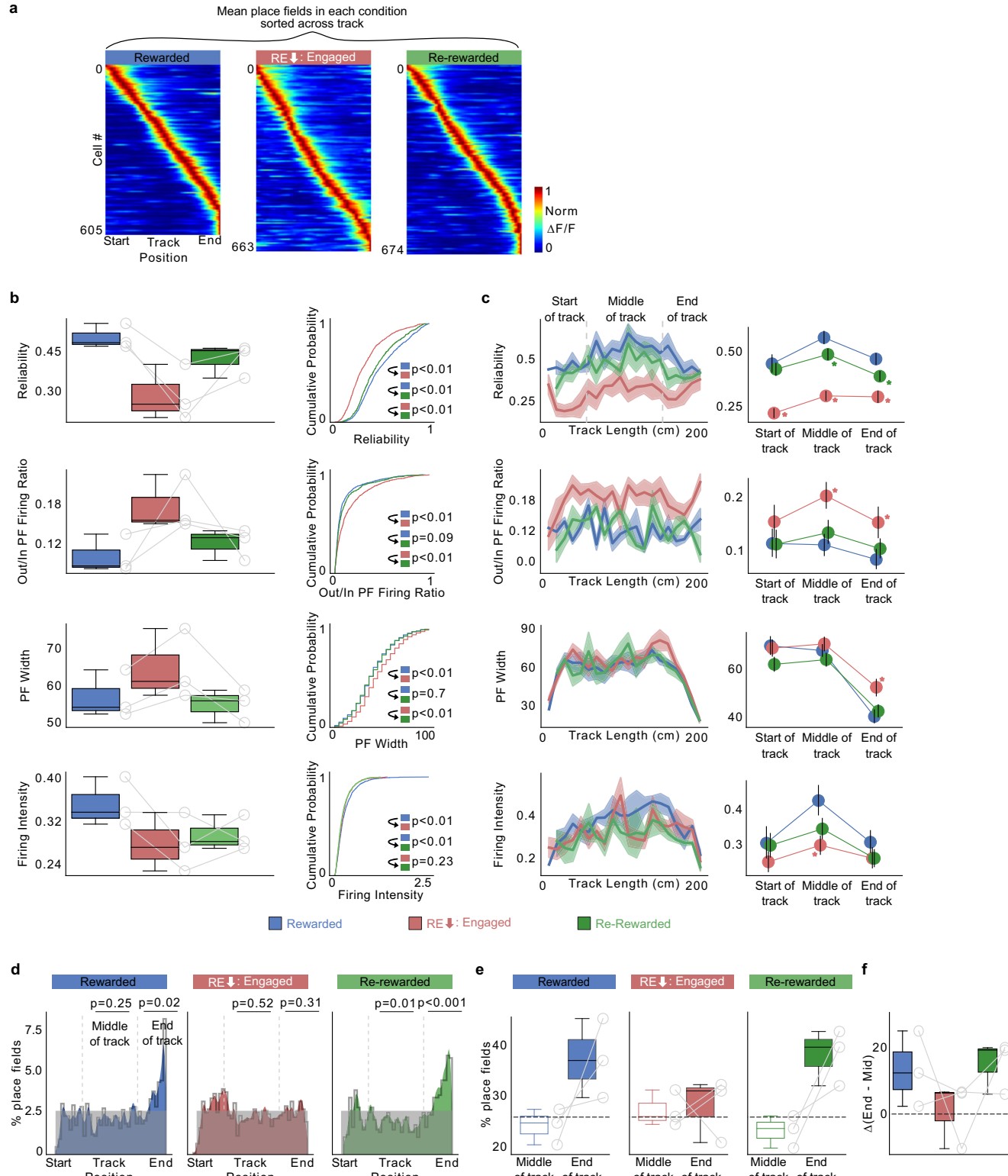

## Lowering reward expectation eradicates the over-representation of reward location and disrupts reward cell firing

One of the striking features of CA1 place cells is the accumulation of place fields near learned rewarded locations[25–29]. We similarly found an over-representation of place fields near the reward site in R (Fig. 3d–f left, l; Supplementary Fig. 11C–E). Interestingly, this over-representation disappeared once reward expectation diminished (Fig. 3d, e middle, f; Supplementary Fig. 11C–E). Of the place cells in R that disappeared in RE$_{low}$, 41.8% (104/249, $n = 3$) were at the end of the

track (150–200 cm). Furthermore, the new place cells that appeared in RE$_{low}$, were equally distributed on the track length (0–50 cm: 77/296 (26%), 50–100 cm: 83/296 (28%), 100–150 cm: 62/296 (21%), 150–200 cm: 74/296 (25%)). When reward was reinstated in RR, the over-representation of the reward site reappeared (Fig. 3d, e right, f; Supplementary Fig. 11C–E). However, not all the place fields that disappeared around the reward zone in RE$_{low}$ reappeared in RR (61/104, 58.6% reappeared), instead, the increased density around the reward site was also derived from the new place cells that formed fields in RR

**Fig. 3 | Diminished reward expectation leads to inferior spatial encoding by unreliable place cells across the entire environment. a** Place fields defined and sorted in each condition pooled from all mice ($n = 3$ mice). Each cell's activity normalized to its peak and cells are sorted by their center of mass along the track. **b** Place cell parameters calculated independently from each condition are displayed as a boxplot of average per animal (left), cumulative histogram (right). *P* values were calculated using a two-sided *t* test. **c** (left) Mean place cell parameters across track location. Shading indicates s.e.m. (right) Average correlation binned by track position indicated by gray lines in the left panel. S: Start of the track, M: Middle of the track, E: End of the track. Asterisk (*) indicates significant *P* values (two tailed KS-test, $P < 0.01$) obtained by comparing R (blue) with other tasks at

each position. **d** Distribution of place field center of mass (COM) locations in each condition pooled from all mice ($n = 3$ mice). Plots show observed density (gray line), uniform distribution (gray shade) and Gaussian distribution of place field density (color). *P* values (two-sided *t* test) were obtained by calculating the place field distribution with the uniform distribution. **e** Percentage of place fields in the middle of the track versus end of the track in each animal (circles). **f** Difference between end of track and middle of track place field percentages in each animal (circles, $n = 3$ mice). Dashed line in **e**, **f** indicates the percentage expected from a uniform distribution across the track. All place field calculations in $RE_{low}$ condition were done on Engaged laps ($RE_{low}$: Engaged). Number of cells in **b–d**; R: 605, $RE_{low}$ Engaged: 663, RR: 674.

(0–50 cm: 31/140 (22%), 50–100 cm: 24/140 (17%), 100–150 cm: 18/140 (13%), 150–200 cm: 67/140 (48%)). Thus, place fields near reward sites act like typical place fields, i.e., they are largely context-specific. Such changes were not observed in control mice (Supplementary Fig. 12). This demonstrates that transitioning from $RE_{high}$ to $RE_{low}$ abolishes place field over-representation of the previously rewarded site. Reinstatement of reward expectation (as in RR) restores over-representation of the reward site with an overlapping yet distinct ensemble of place cells.

It was recently shown that a small fraction of cells distinct from place cells exist in the CA1 that encode reward regardless of position or environment[24]. We looked for these "reward cells" and defined them based on their reward activity in 2 distinct VR environments (Supplementary Fig. 14). We found that these reward cells did not account for the over-representation of the reward site by place cells in R (Supplementary Fig. 14B). We also found that the correlation of reward cell activity between R and $RE_{low}$ was significantly less than within R or between R and RR (Supplementary Fig. 14C, D). Importantly, reduced correlation in reward cell activity was only observed in $RE_{low}$ and not in $RE_{high}$ laps in UR (see example cells in Supplementary Fig. 14D). These findings show that transitioning from $RE_{high}$ to $RE_{low}$ disrupts the additional coding of specific reward sites by place and reward cells.

## Bilateral VTA inhibition largely replicates lowering reward expectation

Ventral Tegmental Area (VTA) dopaminergic inputs to the hippocampus have been implicated in shaping and stabilizing spatial representations[34,37] and VTA/dopamine encode changes in reward expectation[46]. We therefore hypothesized that inhibiting VTA dopaminergic neurons would mimic the effects of lowering reward expectation. To do this, we bilaterally injected Cre-dependent AAV expressing the inhibitory DREADD receptor hM4D(Gi)[47] and mCherry in VTA of DAT-Cre mice and imaged from dorsal CA1 cells expressing GCaMP6f (Fig. 4a, b). On experimental day, mice ran in R for 10 min before being removed and injected intraperitoneally either with saline (control) or one of two different ligands for the hM4D(Gi) receptor— Descholoroclozapine (DCZ)[48] or Clozapine-*N*-oxide (CNO)[47]. Due to the slower kinetics and known off-target actions of CNO, DCZ was also used to inactivate VTA DA neurons. After the injections (45 mins after CNO injections and 10 mins after DCZ injections due to the faster metabolism of DCZ), mice were placed back in R for 10–20 min. Each mouse after training went through 4 days with imaging: Day1: R-UR-RR switch; Day2: Saline session, Day3: CNO, Day4: DCZ session (Fig. 4c). The same FOV was imaged throughout all days of imaging and place cells were extracted from each imaging session (see "Methods"). This protocol allowed us to compare the effect of lowering reward expectation and VTA inhibition on hippocampal neural activity in the same mice.

We found that inactivation by both DCZ and CNO yielded similar results (Fig. 4 and Supplementary Fig. 17), demonstrating a shared mechanism of action and the timing differences following CNO and

DCZ injections and exposure to R, 45 mins versus 10 mins, respectively, does not affect the results. DCZ and CNO administration caused a decrease in lap running speed in R (Supplementary Fig. 16, Mean speed (m/s) [95% CI]: Before Saline 42.74 [41.12 44.36], After Saline 42.75 [41.34 44.17], Before DCZ 41.99 [40.26 43.73], After DCZ 25.96 [24.30 27.62], Before CNO 43.18 [41.26 45.10], After CNO 25.71 [24.03 27.39]) but approach behavior (demonstrating engagement) and anticipatory licking remained (Supplementary Fig. 16). We then measured spatial correlation of place fields before and after DCZ/CNO administration and found a reduction at all locations across the track, similar to the effects in $RE_{low}$ (Fig. 4d–g and Supplementary Fig. 17A–C). This decrease was not observed in the saline control (Fig. 4d–g). To test whether these changes might be due to decreased lap speed after DCZ/CNO administration, we split our saline control data into fast velocity and slow velocity laps (Supplementary Fig. 18A). We found a small reduction in spatial correlation on the slow velocity laps compared to the fast laps (Supplementary Fig. 18B, C), but this was much smaller than the effects induced by DCZ and CNO (Fig. 4d–g and Supplementary Fig. 17A–C).

Interestingly, we found a decrease in lap velocity in control mice following injections (CNO injected in mice with no DREADD expression; Supplementary Fig. 15C, $P < 0.001$). However, this decreased lap speed did not lead to the same changes in place cells we observed in our experimental groups (Supplementary Fig. 15D, E), further indicating that decreased running speed is not the cause of place cell changes we report in our experimental groups injected with CNO/DCZ with DREADD expression in VTA DA neurons (Fig. 4d–g; Supplementary Fig. 17A–C).

We next measured the lap-by-lap reliability of place fields and the out of field firing ratio, two properties most affected by $RE_{low}$. We found a similar decrease in place field reliability and increase in out-of-field firing in DCZ and CNO as in $RE_{low}$ (Fig. 4h, j, Supplementary Fig. 17D, Mean [95%CI]: Reliability: Rewarded = 0.41 [0.31 0.52], $RE_{low}$ = 0.29 [0.17 0.40], Before DCZ = 0.48 [0.38 0.58], After DCZ = 0.41 [0.31 0.51], Before CNO = 0.42 [0.31 0.52], After CNO = 0.35 [0.24 0.47]. Out/In Field Firing: Rewarded = 0.10 [0.06 0.13], $RE_{low}$ = 0.17 [0.06 0.29], Before DCZ = 0.10 [0.05 0.14], After DCZ = 0.12 [0.08 0.16], Before CNO = 0.10 [0.05 0.15], After CNO = 0.12 [0.07 0.17]). This was not the case in the saline controls (Fig. 4i, Mean [95%CI]: Reliability: Before Saline = 0.45 [0.34 0.55], After Saline = 0.46 [0.39 0.54], Out/In Field Firing: Before Saline = 0.09 [0.04 0.14], After Saline = 0.09 [0.04 0.14]), even in the slow velocity saline control laps (Supplementary Fig. 18D, Mean [95% CI]: Reliability: Before Saline_fast velocity = 0.51 [0.46 0.57], After Saline_slow velocity = 0.45 [0.32 0.58], Out/In Field Firing: Before Saline_fast velocity = 0.09 [0.04 0.13], After Saline slow velocity = 0.09 [0.04 0.14]). Lastly, we found DCZ and CNO tended to induce a reduction in place field over-representation of the reward site compared to saline controls, but this did not reach significance (Fig. 4m; $P = 0.06$) and the effect size was less than $RE_{low}$ (Fig. 4k–m and Supplementary Fig. 17E), suggesting other neuromodulators might be involved in over-representation.

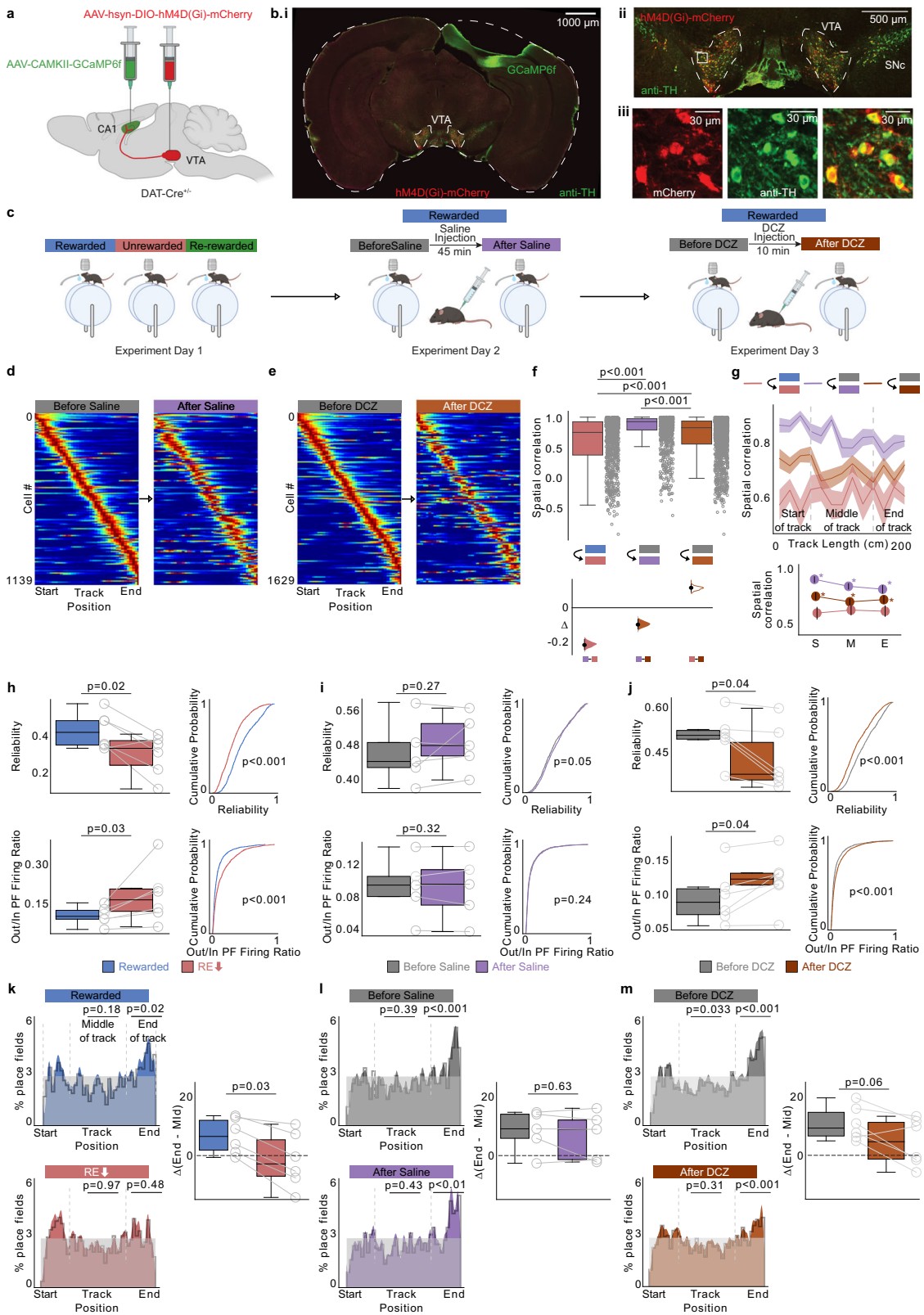

Additionally, both the reduction in over-representation and the changes in reliability and out-of-field-firing ratio were not observed in control animals expressing tdTomato in VTA DA neurons instead of DREADDs and injected with CNO and DCZ (Supplementary Fig. 15). Together, these data indicate that the effects on CA1 spatial encoding with changing reward expectation are largely driven by VTA dopaminergic inputs.

## VTA inputs to CA1 encode reward expectation through a proximity to reward signal

To further investigate how the VTA regulates CA1 encoding, we recorded from VTA dopaminergic (DA) axons directly in CA1 using 2-photon imaging of axon-GCaMP7b specifically expressed in VTA DA axons of DAT-Cre mice (Fig. 5a). We found that individual DA axons ramped up in activity as mice moved closer to the reward site on each

**Fig. 4 | Bilateral inhibition of VTA dopaminergic neurons largely replicates the effects of low reward expectation on place cells. a** Schematic representation of procedure, created with BioRender.com. **b** i, ii: Representative coronal brain section from 1 of 6 mouse brains expressing hm4D(Gi)-mCherry (red) in VTA, GCaMP6f in dorsal CA1(green), and immunostained for Tyrosine Hydroxolase (TH- green). iii: hm4D(Gi)-mCherry expression (left), TH expression (middle) and overlapping expression (right) in example VTA neurons. **c** Experimental protocol. **d**, **e** Place fields defined in the Before Saline/Before deschloroclozapine (DCZ) condition and plotted across After Saline/After DCZ administration. Activity of each place cell was normalized to peak in the Before conditions and sorted by their center of mass. **f** (Top) Boxplots show distribution of place field spatial correlation (circles) in R/ $RE_{low}$ (left, $n = 6$ mice), Before Saline/After Saline (middle, $n = 5$ mice) and Before DCZ/After DCZ (right, $n = 6$ mice). Place cells were defined in the former condition. $P$ values were obtained using two tailed KS-test. (Bottom) Bootstrapped mean differences ($\Delta$) with 95% CI (error bar). **g** (top) Same data, averaged by track traversal of the environment in R (Fig. 5b–d; $n = 7$ axons from 6 mice).

position. Shading indicates s.e.m. (bottom) Average correlation binned by track position indicated by gray lines in the top panel. S: Start of the track, M: Middle of the track, E: End of the track. Asterisk (*) indicates significant $P$ values (two tailed KS-test, $P < 0.01$) obtained by comparing UR (red) with other tasks at each position. **h–j** Place cell parameters in each condition are displayed as boxplot of average per animal (left) and cumulative histogram (right, $P$ values, two-sided paired $t$ test). **k–m** (left) Distribution of place field center of mass (COM) locations in each condition pooled from all mice. Plots show observed density (gray line), uniform distribution (gray shade) and Gaussian distribution of place field density (color). $P$ values (two-sided $t$ test) were obtained by calculating the place field distribution with the uniform distribution. (right) Difference between end of track and middle of track place field percentages in each animal (circles). Dashed line indicates the difference expected from a uniform distribution across the track ($P$ values, two-sided paired $t$ test). Number of cells in **f–g** and **k–m**; R/$RE_{low}$: 928, Before Saline/After Saline: 1139, Before DCZ/After DCZ: 1629.

The activity of these ramping DA axons peaked right before the reward site and then rapidly returned to baseline levels after reward was received (Fig. 5c, d). DA-axon-ramps decayed in slope and amplitude when mice were switched to UR and disappeared in $RE_{low}$ (Fig. 5b–d). DA-axon-ramps started to return early in RR and were almost back to R levels late in RR (Fig. 5e, h). To further quantify DA-axon-ramp dynamics, we measured the slope and max peak of the ramps on each lap throughout R-UR-RR from 5 axons that were imaged throughout all conditions (Fig. 5i–l and Supplementary Fig. 19). We found DA-axon-ramps were consistent in R but decayed abruptly after a few laps in UR (Fig. 5j and Supplementary Fig 19). Similarly, average pre-licking continued for a few laps before rapidly dropping, indicating DA axon-ramps are impacted by reward expectation. DA-axon-ramps were on average smaller in UR compared to R during $RE_{High}$ laps but disappeared in $RE_{low}$ laps (Fig. 5k; $n = 7$, Mean [95%CI]: Slope*max: R = 1.00 [1.00, 1.00], $RE_{High}$ = 0.57 [0.40, 0.74], $RE_{low}$ = 0.24 [0.03, 0.45]). DA-axon-ramps were not different in R versus RR (Fig. 5k; $n = 5$, Mean [95%CI]: Slope*max: R = 1.00 [1.00, 1.00], RR = 1.23 [−0.28, 2.74]). These data demonstrate that VTA DA axons in CA1 encode the animal's proximity to reward and disappear when rewards are no longer expected.

To investigate the emergence of DA-axon-ramps with learning, we switched a subset of mice to a novel (N) VR environment while continuously imaging VTA DA axons ($n = 5$ axons from 5 mice; Supplementary Fig. 20). We found DA axon activity had much lower peaks in N and were much more locked to reward delivery rather than ramping with proximity to reward (Supplementary Fig. 20B, C). However, activity peaks increased with experience in N and DA-axon-ramps started to develop towards the end of the session (Supplementary Fig. 20D), revealing DA-axon-ramps are a learned signal requiring repeated environment-reward associations.

We found two axons with distinct types of signals that did not ramp to reward (Supplementary Fig. 21). One of these encoded the animal's velocity and was not sensitive to the R-UR transition (Supplementary Fig. 21A–C). The other responded to being in VR environments by decreasing activity relative to being in a dark environment (Supplementary Fig. 21D). These findings show that VTA DA axons in CA1 predominantly encode the animal's proximity to reward, but there exists heterogeneity across the population of DA axons with some axons encoding other features of experience[35].

## Discussion

During wakeful exploration animals continuously experience external events, some of which are robustly encoded into memory for future recall. A key aspect of whether external events become encoded into memory depends on the internal state of the animal during encoding[11,49]. Here we found that changes in reward expectation within unchanging spatial environments alters the structure and trial-to-trial dynamics of place codes in CA1 likely through the modulation of a ramping to reward signal in dopaminergic inputs from VTA to CA1. This is supported by several observations: 1, reward removal led to diminished reward expectation which caused an abrupt restructuring of the place code that included place cell remapping, the loss of some place fields, and the formation of new place fields at all locations within the environment, plus a loss of place field over-representation of the reward zone. 2, place code restructuring only occurred after the reward expectation diminished and not following reward removal 3, place cells encoding the environment during low reward expectation were degraded in quality exhibiting low trial-to-trial reliability and high out-of-field firing at all locations 4, bilateral inhibition of dopaminergic neurons in the VTA during high reward expectation largely mimicked the effects of lowering reward expectation 5, dopaminergic axons from the VTA to CA1 encoded a ramping to reward signal during high reward expectation that disappeared after lowering reward expectation. These results provide evidence that the structure and robustness of spatial memory encoding in the hippocampus is determined by dopaminergic inputs from the VTA that is dependent on the animal's internal state of reward expectation during navigation.

Reward contingencies in navigation tasks have been shown to modulate place cells[23,24,26,33,50]. Most studies in this area either alter reward magnitudes or move reward locations, and many times include a decision-making component in their behavioral task. These factors all modulate place cells in some way, depending on the specifics of the experiment. What has been difficult to achieve in this research area is a complete removal of rewards for long enough to alter, and measure, reward expectation yet maintain matched navigation behavior. This is a necessary step to assess the influence of changing reward expectations on place cells without confounds caused by changes in navigation behavior[39]. Our behavioral set-up allowed us to match navigation behaviors, even when reward was not expected. Specifically, head direction, location occupancy, location sequences leading to reward, running speed, and pupil area were the same in rewarded and unrewarded conditions for many trials. A number of factors led to this matched behavior: (1) Mice were head-fixed; (2) The behavior was simple and stereotyped (mice run on a linear treadmill along a linear track); (3) Mice were first trained to run to a very high level with reward to establish high reward expectation before reward was removed; (4) Many traversals of the environment could be achieved in short succession (~5 traversals/min). In conjunction with our ability to measure reward expectation on a trial-by-trial basis, this matched behavior allowed us to specifically connect the influence of reward expectation on place cells in real-time.

Interestingly, our data show that the presence (or consumption) and subsequent absence of reward itself has little influence on spatial encoding in CA1 as shown by very little difference in the spatial code between R and UR when RE is high on the first few laps. Sharp wave

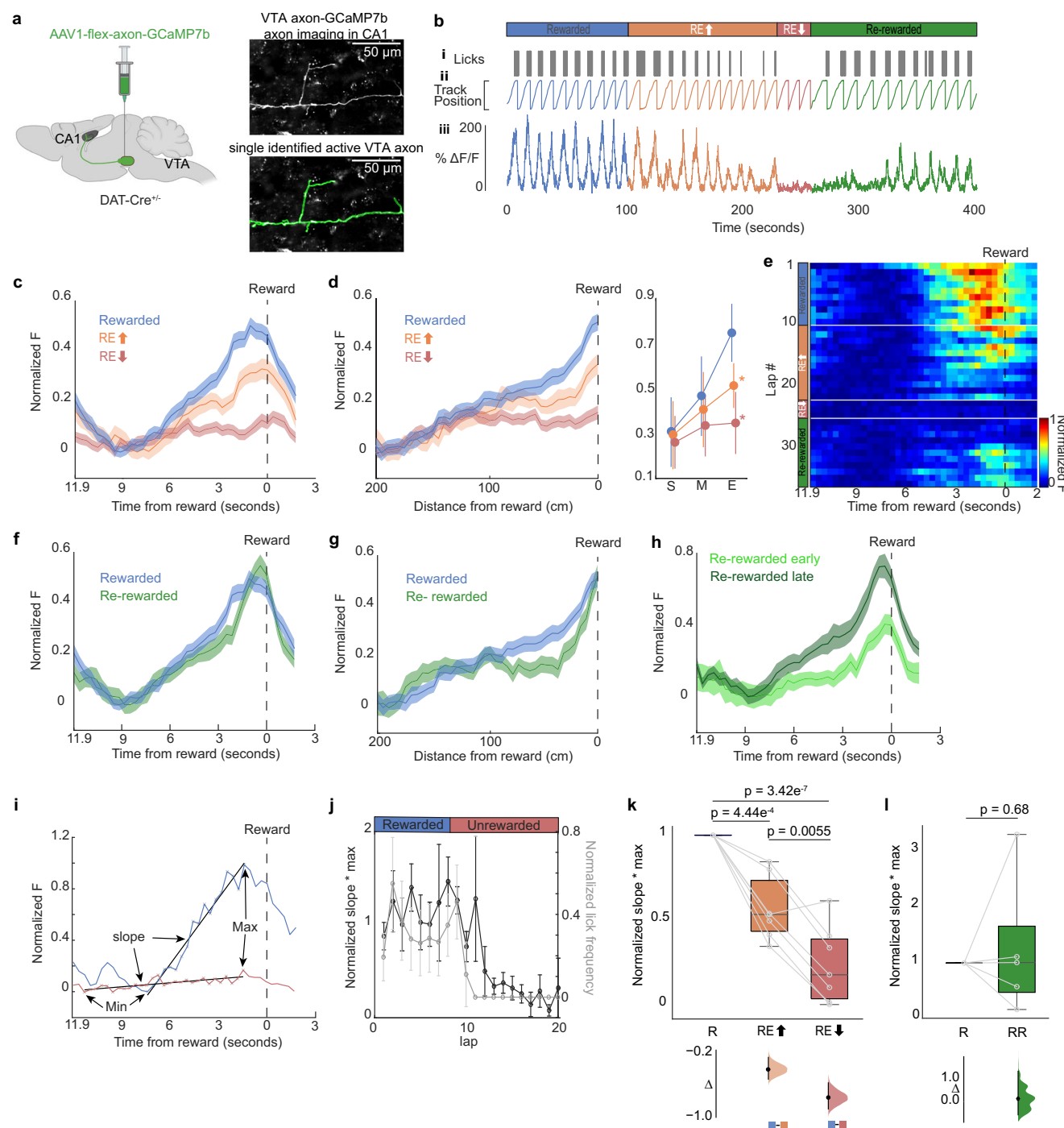

**Fig. 5 | Activity of dopaminergic VTA axons in CA1 ramp up to reward.**
**a** Schematic representation of injection procedure created with BioRender.com (left). Example CA1 field of view of VTA axons (right, top). Extracted region of interest (right, bottom). **b** Example mouse. i: Mouse licking behavior. ii: Mouse track position. iii: ΔF/F from an example ROI. **c** Fluorescent activity of axons (7 axons in 6 mice) in R (blue), $RE_{high}$ (orange-RE arrow), and $RE_{low}$ (red-RE arrow) experimental conditions averaged by time to reward. Shaded areas represent s.e.m. **d** Same data, averaged by position (left). Mean with 95% CI (error bar) of starting 50 cm (S), middle 100 cm (M), and end 50 cm (E)(right). Asterisk (*) indicates significant p-values (two-sided paired $t$ test, $P < 0.01$) obtained by comparing R (blue) with other tasks at each position. **e** Normalized fluorescence of an example axon in the different conditions binned by time to reward. White lines divide each condition, and the dashed line represents time of reward delivery. **f** Fluorescent

activity of VTA axons (5 axons in 5 mice) in R(blue) and RR (green) averaged by time to reward. Shaded areas represent s.e.m. **g** Same data averaged by position. **h** RR time binned fluorescent activity divided into early (light green) and late laps (dark green) and averaged by time to reward. **i** Example showing how the max and slope of time binned fluorescence data was determined in an R (blue) and $RE_{low}$ (red) lap. **j** Mean slope*max for laps in the R and Unrewarded (black, $n = 7$) and mean lick frequency normalized to maximum licks (gray), error bars represent s.e.m. **k, l** Boxplot shows distribution of mean slope*max of axons (circles) within R, $RE_{high}$, and $RE_{low}$ (**k**; $n = 7$ axons) and within R and RR (**l**; $n = 5$ axons). $P$ values were obtained using a two-sided paired $t$ test. (Bottom) Bootstrapped mean differences (Δ) with 95% CI (error bar) are shown at the bottom. $X$-axis indicates the comparisons made.

ripples and place cell replay events occur more frequently during reward consumption which might reflect a reward-related feedback signal that could influence the place code for the environment, but this does not seem to be the case[51,52]. Instead, it is only after animals learn to associate or disassociate reward from the environment and change their reward expectation that we observe changes in the place cell code. The structure of the map, place cell over-representation of the reward location, and trial-by-trial reliability/out-of-field firing were all modulated only when reward expectation changed, not when reward was removed/added. This demonstrates that the act of reward attainment does not in itself modulate place cells in the hippocampus, which may have been the case through a reward-related feedback signal. Instead, the animal's internal state of reward expectation is a stronger driver of place cell encoding than the external reward. Reward cells in this region have recently been described that encode reward independently of space[24]. We did identify these cells, and found that they too were modulated by reward expectation rather than reward per se. Therefore, our data suggests internal states of reward expectation rather than reward attainment modulate hippocampal spatial encoding of locations within environments.

Such changes in the place code in an unchanging spatial environment could reflect the animal's attempts to infer whether they are in a different "state" of the world[53], something that has been observed in the prefrontal cortex[54,55]. The sensory cues remained constant but internal expectation of reward did not, and this was sufficient to induce partial remapping along with other changes in the place code. From the perception of the mice, the state of the world changed as the locations within the environment became devalued by lowering reward expectation and no longer predicted the presence of a reward. The *value* of the sensory experience thus alters the CA1 place code, suggesting that the hippocampus does not simply represent spatial information, but flexibly encodes the value of space and is able to discriminate contexts within an unchanging spatial environment[53]. This could be due to mice disengaging from their environment once it is no longer valued. A recent paper showed that when mice disengage with their environment CA1 place codes degrade[41]. However, we measured two distinct features—approach behavior and pupil area—as a readout of engagement and found mice in our experiments remain engaged on many trials even when reward expectation is low. For instance, mice decelerated as they approached the end of the track on many trials even when reward expectation had diminished. This approach behavior was likely due to mice anticipating hitting the VR wall at the end of the track which triggers a 2 s delay before teleportation to the start of the track. This was not due to stereotyped behavior as switching mice to a dark environment with no spatial cues eradicated approach behavior. Therefore, mice in our set up were engaged with their spatial environment following lowered reward expectation on many trials. The changes in the place code we observe are therefore unlikely due to disengagement with the environment and are instead most likely due to reduced reward expectation. Although our mice show signs of engagement, we cannot rule out that they are less engaged than during high reward expectation and our engagement measures are not sensitive enough to capture more subtle changes. Reduced engagement could therefore contribute to the place cell changes we report here. In agreement with Pettit et al.[41], we did find obvious disengaged trials during low reward expectation and these trials show further place code changes beyond those caused by reduced reward expectation. We therefore add to the Pettit et al.[41] findings by demonstrating extensive changes in CA1 place coding caused by lowered reward expectation during engagement (or minimally lowered engagement) with the environment. This implicates additional internal states in modulating place coding beyond environment engagement-disengagement.

The process of switching reward expectation from high-low-high led to a greater distinction between representations in the two high reward expectation conditions than when reward expectation was held high throughout the session (as was the case in control animals that maintained high reward expectation throughout a time-matched session). This suggests the CA1 chunks external events into distinct episodes based on changes in internal state, even when external events remain the same. In other words, when internal expectations are constant, unchanging external events are encoded as a single contextual episode. When internal expectations change and then return to original levels, unchanging external events are encoded as distinct contextual episodes. This encoding of episodic information within the CA1 network is consistent with its proposed role in capturing temporal and contextual episodes[56].

DA activity in CA1 is known to play an important role in hippocampal-dependent reward learning[57] and DA VTA inputs to CA1 have been shown to modulate reward learning[34]. Optogenetic activation of VTA axons during learning of new goal locations enhances the subsequent reinstatement of spatial representations and stabilizes memory performance[34]. The stability of CA1 spatial representations is reduced by inactivation of VTA neurons[33] or CA1 DA receptor antagonism[58]. Here, we also find that spatial representations are more stable when VTA DA neurons are active (during high reward expectation) and destabilize when VTA DA activity is reduced (during lowered reward expectation and during VTA DA inhibition). What is missing from previous studies is a direct connection between the animal's internal state of reward expectation and how changing reward expectation changes spatial representations and VTA DA activity. Our findings fill this gap and we additionally reveal that bilateral inhibition of VTA DA neurons: 1, decreases the trial-to-trial reliability and increases the out-of-field firing of place fields. 2, reduces reward site over-representation by place fields. 3, largely replicates the effects of lowering reward expectation. We also reveal the natural VTA DA dynamics in CA1 and their response to changes in reward expectation (further discussion below), which, as far as we know, has never been measured. Our findings therefore replicate previous work on VTA DA influences on spatial representations and memory but add to this area by showing the natural dynamics of VTA DA inputs to CA1 and their influence on spatial representations in real-time during changes in reward expectation. We also show the "fate" of individual place cells through changes in reward expectation, revealing heterogeneity of responses at the single cell level (some place fields disappear when reward expectation is lowered, some remap, some cells form new place fields, and some maintain their place fields). One caveat is that VTA inhibition did cause lap velocity to reduce - a known role of VTA dopaminergic neurons[58]—which itself could cause changes to place coding[39]. We addressed this by comparing slow and fast velocity laps in saline controls to see if place cell differences could be observed, and we did find small differences. Therefore, reduced velocity likely contributes to the observations we made during VTA inhibition but given the small effect size of velocity it is unlikely to explain all the changes. Our findings instead support a framework whereby diminished reward expectation causes diminished DA release from VTA in CA1, leading to an abrupt restructuring of place coding that includes a loss of over-representation of rewards sites, plus a degradation in the quality of place coding.

In support of this idea, we show that VTA DA neurons in the hippocampus exhibit ramping to reward activity that diminishes following the removal of rewards. Similar activity has been observed in VTA DA neurons and their projections to several brain areas using various techniques[30,36,38,59]. It is not entirely clear whether this ramping activity signals reward prediction error (RPE)[36] or value[30,59]. However, our findings that VTA axon activity peaks *prior* to expected reward locations in familiar environments but peaks *at* the location of unexpected rewards in novel environments supports the established idea that VTA DA neurons signal RPE. Importantly, this ramping activity diminished over the course of several trials following reward removal, was

completely absent in RE$_{low}$, and rapidly re-established in RR mimicking the timing of changes to reward expectation and hippocampal place cell codes we observed. Thus, we hypothesize that this ramping activity provides reward expectation information to the hippocampus through DA release that is required to maintain specific excitatory drive to place cells or postsynaptic responses within place fields. However, we did not specifically manipulate this input in our DREADD experiment, which targeted all DA VTA neurons and not just DA inputs to CA1. Future experiments should be designed to specifically manipulate the VTA-CA1 DA input to further test our hypothesis.

Interestingly, we found that although our CNO/DCZ inhibition experiments largely replicated the results of lowering reward expectation, it did not cause a corresponding inhibition of pre-licking. This suggests a distinct brain region might encode reward expectation and send parallel signals to both VTA, to drive DA ramps, and to a lower-order center that drives pre-licking. In our CNO/DCZ experiment we inhibited the VTA component of this circuit which appears to leave intact the parallel circuit to licking centers. Indeed, there is evidence for such a reward expectancy center in PFC[60].

Although VTA DA inputs to CA1 are sparse, manipulation of this pathway has large effects on spatial memory[34,61,62] and many subtypes of DA receptors are expressed throughout CA1 on pyramidal cells, Interneurons, and astrocytes[63,64]. The influence of this sparse input could be amplified by the types of connections VTA DA inputs make. A recent paper showed that VTA DA inputs to Nucleus Accumbens make "spinule" connections that increase the surface area between DA inputs and their postsynaptic targets, potentially amplifying their influence[65]. Volume transmission is another potential mechanism that could amplify DA's influence on the hippocampus[63]. Local interneurons and/or astrocytes expressing DA receptors could further amplify DA's influence in CA1 through their many connections with pyramidal cells. While we did not measure DA release in the hippocampus and cannot directly attribute the effects of bilateral VTA DA neuron inhibition to DA activity, studies have demonstrated that these neurons can impact hippocampal place cell stability through DA receptor dependent mechanisms[34]. DA regulates synaptic transmission and dendritic excitability[66] and high dendritic excitability of CA1 basal dendrites has been linked to place field emergence, precision, and long term stability[44,67]. Interestingly, VTA DA inputs are located in the Stratum Oriens of CA1 where basal dendrites of pyramidal cells reside and express D5 receptors[68]. We hypothesize that DA increases dendritic excitability to increase dendritic branch spike prevalence across basal dendrites when reward expectation is high. High dendritic branch spike prevalence stabilizes place fields and increases their precision and reliability[44,67]. Following diminished reward expectation, loss of DA reduces branch spike prevalence, destabilizing place fields (restructuring the place code at the population level) and reducing their trial-to-trial precision and reliability. This hypothesis remains to be tested.

Although the presence of DA-ramps in CA1 means the level of DA release is not equal across the track, attractor-like dynamics could ensure DA influences place cells at all locations. For instance, place cells with place fields closer to the end of the track receive greater levels of DA release, but are part of a larger place cell sequence (possible attractor network) with cells that have place fields at the beginning of the track that receive lower levels of DA. Based on the known connectivity of CA1 neurons, this may arise from local inhibition within CA1 or driven by input from CA3, which does have recurrent connectivity to support attractor dynamics and also receives dopaminergic input[32,57,69]. Furthermore, place cells sequences are replayed during immobility and reward enhances the fidelity and increases the frequency of replays which could further stabilize place cell sequences associated with high reward expectation that include place fields throughout the entire track and not just ones close to the reward site[52,70,71]. A hypothesis generated from this framework would be that

DA ramps increase replay fidelity and/or frequency. Indeed, optogenetic stimulation of DA neurons in VTA enhances replay events in CA1, suggesting the DA ramps we observed may be the natural brain signal that leads to a similar enhancement of CA1 replay events.

An alternative source of dopamine in CA1 could be coming from locus coeruleus (LC) fibers which impact hippocampal learning and memory in a DA dependent manner[72]. LC has been shown to encode reward expectation[73] and a recent study found optogenetic stimulation of LC-CA1 inputs at a goal induced a shift in place fields towards the goal, whereas inhibition decreased overrepresentations of new goal locations suggesting these inputs help establish over-representation of reward locations[27]. However, these inputs only showed activity locked to new goal locations but not familiar locations and have not been shown to influence pre-existing overrepresentation of goal locations or place fields throughout an environment. Therefore, it is unlikely that these inputs are the main driver for restructuring of the place code observed during diminished reward expectation. It is possible LC inputs do have some influence, though, as inhibition of VTA by DCZ or CNO did not induce the same effect size as lowering reward expectation, implicating other neuromodulatory systems beyond VTA. Serotonergic inputs to the hippocampus from the Raphe Nuclei also encode reward related information, so could further modulate CA1 place codes during changes in reward expectation[74]. Given the importance of strongly encoding reward-related memories it is not surprising that reward-related information is distributed across multiple neuromodulatory systems that project to the hippocampus. However, our findings suggest VTA DA is the main system in modulating dorsal CA1 during changes in reward expectation during spatial navigation.

An outstanding question is what drives the DA ramps in VTA. Ramps seem to require animals to know where they are and how far they are from a reward. This implicates the hippocampus and the place codes represented there in providing spatial information to the VTA[38]. It has been proposed that a loop may exist between the hippocampus and VTA whereby the hippocampus sends information through the subiculum, accumbens, and ventral pallidum to the VTA[57]. This pathway could inform DA neurons in VTA of the animal's position relative to reward. These neurons could then ramp up their firing as the animal approaches expected reward locations, modulating the value of locations based on their distance from reward locations. The DA released in the hippocampus thus serves to stabilize the structure of place codes and maintain reliable place fields along trajectories that lead to expected rewards[38].

## Methods
### Subjects
All experimental and surgical procedures were in accordance with the University of Chicago Animal Care and Use Committee guidelines. For this study, we used 10–12-week-old male C57BL/6J wildtype (WT) mice and *Slc6a3*$^{Cre/+}$ (DAT-Cre$^{+/-}$) mice (23–33 g). Male mice were used over female mice due to the size and weight of the headplates (9.1 mm × 31.7 mm, ~2 g) which were difficult to firmly attach on smaller female skulls. Mice were individually housed in a reverse 12 h light/dark cycle at 72 °F and 47% humidity, and behavioral experiments were conducted during the animal's dark cycle.

### Mouse surgery and viral injections
Mice were anesthetized (~1–2% isoflurane) and injected with 0.5 ml of saline (intraperitoneal injection) and 0.5 ml of Meloxicam (1–2 mg/kg, subcutaneous injection) before being weighed and mounted onto a stereotaxic surgical station (David Kopf Instruments). A small craniotomy (1–1.5 mm diameter) was made over the hippocampus (1.7 mm lateral, −2.3 mm caudal of Bregma). For population imaging, a genetically-encoded calcium indicator, AAV1-CamKII-GCaMP6f (pEN-N.AAV.CamKII.GCaMP6f.WPRE.SV40 was a gift from James M. Wilson –

Addgene viral prep #100834-AAV1; https://www.addgene.org/100834/; RRID:Addgene_100834) was injected (-50 nL at a depth of 1.25 mm below the surface of the dura) using a beveled glass micropipette leading to GCaMP6f expression in a large population of CA1 pyramidal cells. For DREADD experiments, craniotomies were made over the hippocampus and bilaterally over the ventral tegmental area (VTA) (±0.5 mm lateral, 3.1 mm caudal of Bregma) of DAT-Cre[+/-] mice. A genetically encoded DREADD receptor (pAAV-hSyn-DIO-hM4D(Gi)-mCherry was a gift from Bryan Roth (Addgene viral prep # 44362-AAV1; http://n2t.net/addgene:44362; RRID: Addgene_44362) or tdTomato (pAAV-FLEX-tdTomato was a gift from Edward Boyden (Addgene viral prep # 28306-AAV1; http://n2t.net/addgene:28306; RRID: Addgene_28306) was injected (-200 nL at a depth of 4.4 mm below the surface of the dura). For axon imaging, a small craniotomy was made over the ventral tegmental area (VTA) (0.5 mm lateral, −3.1 mm caudal of Bregma) of DAT-Cre[+/-] mice. A genetically-encoded calcium indicator, pAAV-Ef1a-Flex-Axon-GCaMP7b (pAAV-Ef1a-Flex-Axon-GCaMP7b was a gift from Rylan Larsen · Addgene plasmid # 135419; http://n2t.net/addgene:135419; RRID: Addgene_135419) was packaged into AAV1 and injected (-200 nL at a depth of 4.4 mm below the surface of the dura) leading to axon-GCaMP7b expression in dopaminergic VTA neurons. Afterwards, the site was covered up using dental cement (Metabond, Parkell Corporation) and a metal head-plate (9.1 mm × 31.7 mm, Atlas Tool and Die Works) was also attached to the skull with the cement. Mice were separated into individual cages and water restriction began the following day (0.8–1.0 ml per day). For axon imaging, mice were put on water restriction 3 weeks after viral injection to provide time for increased expression of axon-GCaMP7b. On the 7th day of water restriction, mice underwent another surgery to implant a hippocampal window as previously described[75]. Following implantation, the head-plate was reattached with the addition of a head-ring cemented on top of the head-plate which was used to house the microscope objective and block out ambient light. Post-surgery mice were given 2–3 ml of water/day for 3 days to enhance recovery before returning to the reduced water schedule (0.8–1.0 ml/day). Expression of GCaMP6f reached a somewhat steady state -20 days after the virus was injected.

## Behavior and virtual reality
Our virtual reality (VR) and treadmill setup was designed similar to previously described setups[44,76]. The virtual environments that the mice navigated through were created using VIRMEn[77]. 2 m linear tracks rich in visual cues were created that evoked numerous place fields in mice as they moved along the track at all locations (Fig. 1a)[78]. Mice were head restrained with their limbs comfortably resting on a freely rotating styrofoam wheel ('treadmill'). Movement of the wheel caused movement in VR by using a rotary encoder to detect treadmill rotations and feed this information into our VR computer, as in refs. 46, 77. Mice received a water reward (4 μL) through a waterspout upon completing each traversal of the track (a lap), which was associated with a clicking sound from the solenoid. Licking was monitored by a capacitive sensor attached to the waterspout. Upon receiving the water reward, a short VR pause of 1.5 s was implemented to allow for water consumption and to help distinguish laps from one another rather than them being continuous. Mice were then virtually teleported back to the beginning of the track and could begin a new traversal. Mouse behavior (running velocity, track position, reward delivery, and licking) was collected using a PicoScope Oscilloscope (PICO4824, Pico Technology, v6.13.2). Pupil tracking was done through the imaging software (Scanbox v4.1, Neurolabware) at 15 frames per sec, using Allied Vision Mako U-130b camera with a 25 mm lens and a 750 nm longpass IR filter. IR illumination from the objective was used to illuminate the pupil for tracking. Behavioral training to navigate the virtual environment began 4–7 days after window implantation (-30 min per day) and continued until mice reached >4 laps per minute, which took 10–14 days (although some mice never reached this

level). This high level of training was necessary to ensure mice continued to traverse the track similarly after reward was removed from the environment. Initial experiments showed that mice that failed to reach this criterion typically did not traverse the track as consistently without reward. Such mice were not used for imaging. The rate of success in training mice to reach this criterion was ~60%. In mice that reached criteria, imaging commenced the following day. Additionally, since we are testing changes in reward expectation, only animals that displayed pre-licking in the familiar environment before reward delivery were used for imaging.

## Two-photon imaging
Imaging was done using a laser scanning two-photon microscope (Neurolabware). Using a 8 kHz resonant scanner, images were collected at a frame rate of 30 Hz with bidirectional scanning through a 16x/0.8 NA/3 mm WD water immersion objective (MRP07220, Nikon). GCaMP6f and GCaMP7b were excited at 920 nm with a femtosecond-pulsed two photon laser (Insight DS + Dual, Spectra-Physics) and emitted fluorescence was collected using a GaAsP PMT (H11706, Hamamatsu). The average power of the laser measured at the objective ranged between 50–70 mW. A single imaging field of view (FOV) between 400–700 μm equally in the $x/y$ direction was positioned to collect data from as many CA1 pyramidal cells or dopaminergic axons as possible. Time-series images were collected through Scanbox (v4.1, Neurolabware) and the PicoScope Oscilloscope (PICO4824, Pico Technology, v6.13.2) was used to synchronize frame acquisition timing with behavior.

## Imaging sessions
The familiar environment was the same environment that the animals trained in. The experiment protocol for single day imaging sessions is shown in Fig. 1a. Each trial lasted -8–12 min and was always presented in the same order. 6 mice were exposed to Rewarded (R), Unrewarded (UR) and Re-Rewarded environments (RR), in that order. An additional 6 mice were exposed to only R and UR. Mice on average ran $34 \pm 2$ (mean ± 95% CI) laps in the Rewarded condition, at which point, reward was turned off and imaging in the Unrewarded environment continued ($28 \pm 4$ laps). In the Unrewarded condition, both reward and auditory cue associated with the reward (solenoid click) were disabled. In $n = 6$ animals, reward was then turned on again (Re-rewarded) and mice ran $27 \pm 3$ laps. To identify reward cells, the 6 mice that went through R-UR-RR were also introduced to a Novel-rewarded environment (NR; $31 \pm 5$ laps). The Novel-rewarded environment (N) had distinct visual cues, colors and visual textures, but the same dimensions (2 m linear track) and reward location (end of the track) as the familiar environment. Furthermore, to rule out the possibility that observed changes in population activity were due to time, mice were exposed to only the familiar Rewarded environment for 20 min (control, $n = 6$).

## DREADD experimental protocol
To activate hM4D(Gi) receptor and silence VTA dopaminergic neurons, two ligands were used - Deschloroclozapine dihydrochloride (DCZ, MedChemExpress)[48] and Clozapine N-Oxide (CNO, Enzo Life Sciences, Inc)[47]. Due to the slow kinetics and known off-target effects of CNO, DCZ was used as an additional method for inactivation[48].

CNO was dissolved in DMSO at a 5 mg/mL concentration and stored at −80 °C. On experiment day, CNO solutions were thawed at room temperature and diluted to 0.6 mg/mL with saline (details on saline). DCZ was dissolved in DMSO at 5 mg/mL concentration and stored at −80 °C. On experiment day, DCZ solutions were thawed at room temperature and diluted to 0.02 mg/mL with saline.

Once DREADD or tdTomato (control) injected DAT-Cre mice met training criteria, they were habituated to the injection process. They were exposed to the familiar rewarded environment for ~10 min. Afterwards, they were removed from the VR set up, placed in the

holding room, and injected with ~150 μL of a 12% DMSO/Saline solution. After ~30–45 min, they were placed back in the VR setup and exposed to the familiar rewarded environment for an additional 10 min. This was repeated for 3–5 days to acclimate mice to the injection procedure.

The experimental protocol for the first day of DREADD experiments was identical to the reward manipulation experiments described above. At the end of the imaging session, a 1 minute time-series movie was collected at a higher magnification and then averaged to aid as a reference frame in finding the same imaging plane on subsequent days. On Experiment Day 2, mice were first exposed to R for 8–12 min. The mice were removed from the VR set up and placed in a holding room where they were immediately injected with ~150 μL of a 12% DMSO/ Saline solution. ~35 min after injection mice were placed back on the VR setup and the same imaging plane was found. At the 45 min post injection mark, mice were again exposed to R for 15–20 min. The procedure for Experiment Day 3 was identical to Day 2 except mice were injected with 5 mg/kg CNO in a 0.6 mg/mL solution instead of the DMSO/Saline solution or with 0.1 mg/kg DCZ of a 0.02 mg/mL solution. Due to the faster kinetics of DCZ, mice were placed back in R after 10 min post injection.

## Histology and brain slices imaging
We checked the VTA expression of hm4D(Gi)-mCherry to confirm adequate coverage of dopaminergic VTA neurons. Mice were anesthetized with isoflurane and perfused with ~10 ml phosphate-buffered saline (PBS) followed by ~20 mL 4% paraformaldehyde in PBS. The brains were removed and immersed in 30% sucrose solution overnight before being sectioned at 30 μm-thickness on a cryostat. Brain slices were collected into well plates containing PBS. Slices were washed 5 times with PBS for 5 min then were blocked in 1% Bovine Serum Albumin, 10% Normal goat serum, 0.1% Triton X-100 for 2 h. Brain slices were then incubated with 1:500 rabbit-α-TH (MAB318, Sigma Aldrich) in blocking solution at 4 °C. After 48 h, the slices were incubated with 1:1000 goat-α-rabbit Alexa Fluor 488 secondary antibody (A32731, ThermoFisher) for 2 h. Brain slices were then collected on glass slides and mounted with a mounting media with DAPI (SouthernBiotech DAPI-Fluoromount-G Clear Mounting Media, 010020). The whole-brain slices were imaged under ×10 and x40 with a Caliber I.D. RS-G4 Large Format Laser Scanning Confocal microscope from the Integrated Light Microscopy Core at the University of Chicago.

## Image processing and ROI selection
Time-series images were preprocessed using Suite2p (v0.10.1)[79]. Movement artifacts were removed using rigid and non-rigid transformations and assessed to ensure absence of drifts in the z-direction. Datasets with visible z-drifts were discarded (n = 2). For multi-day datasets (DREADD Experiments), imaging planes acquired from each day were first motion corrected separately. ImageJ (v1.53, NIH) was then used to align the motion corrected images relative to each other by correcting for any rotational displacements. The images across all days were then stitched together and motion corrected again as a single movie. For population imaging, regions of interest (ROIs) were also defined using Suite2p (Fig. 1a) and manually inspected for accuracy. Baseline corrected ΔF/F traces across time were then generated for each ROI and filtered for significant calcium transients, as previously described[44,67,75]. Finally, we used raster plots[80] to visualize the ΔF/F population activity of neurons across time and across all conditions (Fig. 1c and Supplementary Figs. 2 and 3). In these raster plots, neurons were clustered and sorted such that neurons with correlated activity were next to each other on the vertical axis (https://github.com/MouseLand/rastermap). For visual clarity, only neurons with at least 2 transients above 10% ΔF/F over the time of the experiment were included in the raster plot and the 2-D plots were interpolated using a hanning filter.

For axon imaging, ROIs were first defined using Suite2p and manually inspected for accuracy. ROIs were then hand drawn over all segments of Suite2p defined active axons using ImageJ to ensure all axon segments were included for analysis. Fluorescent activity for each ROI was extracted and highly correlated ROIs (Pearson correlation coefficient ≥ 0.7) were combined and fluorescent activity for the combined ROI was extracted. Baseline corrected ΔF/F traces across time were then generated for each ROI using a larger sliding window of 2000 frames.

## Licking behavior
Licking data was collected using a capacitive sensor on the waterspout. Well trained mice showed a higher proportion of licks (pre-licking) in the region immediately preceding the reward in R (Supplementary Fig. 1). This anticipatory licking behavior continued for a few laps in UR (5 ± 1 lap) and decayed exponentially (Fig. 1e) except for some animals (4/12) that randomly licked in later laps. To calculate anticipatory licking in UR, we defined a reward zone which started from the average track position at which the animal started pre-licking in R and ended after teleportation. We calculated the presence of any licks within this zone to quantify anticipatory licking in UR in the absence of a reward. The lap where pre-licking stops in UR was then defined as the lap following 2 consecutive laps with an absence of these licks.

## Position decoding
We trained a naive Bayes decoder (scikit-learn, v1.0.2, Python[81]) to predict the spatial location of the animal on the linear track from population activity within each mouse. Population activity consisted of ΔF/F traces from all identified cells organized as NxT, where N is number of cells and T is the total number of frames from an imaging session. Each lap traversal on the 2 m track was discretized into 40 spatial bins (each 5 cm wide). Time periods where the animal was stationary were filtered out (speed <1 cm/s) and the decoder was only trained on frames belonging to running periods >1 cm/s. Running behavior and population activity before and after filtering is shown in Supplementary Fig. 3 and Fig. 1c, respectively. To ensure decoder performance was not confounded by teleportation, we considered the end of the track as continuous with the beginning of the track so that the topology of the track was treated as a circle.

To assess how well a decoder trained in R was able to decode the animal's spatial location in other conditions (Fig. 1d–g), the decoder was trained on the first 60% of laps in R. The resulting model was evaluated on the remaining laps in R and on all laps in UR and RR (Fig. 1d). Quality of fit was assessed by calculating the coefficient of determination ($R^2$) between the actual location of the animal and the location predicted by the decoder. Decoder error was quantified as difference in actual and decoded position in cm (Fig. 1g). We also trained and tested decoders within each condition in each mouse (Supplementary Fig. 13). Here, to assess decoder performance and to account for population activity changes across time, we employed a cross-validation approach by sliding the tested laps (20% of laps) by one each time and training on the remaining laps (80% of laps). Furthermore, to account for different numbers of laps across conditions, we down sampled each condition to match the condition with the least number of laps.

## Decoder performance with different behavioral parameters
**Licking behavior.** To analyze the relationship between decoder error and licking, we identified the lap when licking had stopped in UR when 2 consecutive laps had no licks, and then divided the data into laps before licking stopped ($RE_{high}$, before licking stops) and laps after licking stopped ($RE_{low}$, after licking stops). We found that if we instead used different criteria to identify when licking had stopped, i.e., the first lap with no licks, or 4 consecutive laps with no licks, our results were unaffected. This was also true if instead of defining when licking

stopped in UR we simply grouped laps together based on the presence or absence of licks (Supplementary Fig. 4). However, with 6 consecutive laps with no licks, our results differed (Supplementary Fig. 4). We obtained the lap wise decoder fit ($R^2$) and lick frequency in UR in each animal and ran a rolling average with a sliding window of 3 laps (Fig. 1e). The average decoder fit across laps formed an S-shaped curve. We fit this mean $R^2$ to a reverse Boltzmann Sigmoid curve (scipy.curve_fit, v1.7.3, Python, Fig. 1e, coefficient of determination of curve fit with mean decoder = 0.94). To calculate the inflection point at which the rate of decrease in $R^2$ reaches the maximum, we calculated the first point where the second derivative of the fit reached 0 (lap 10, Fig. 1e).

**Time taken to complete a lap.** This was calculated as the total time (in seconds) taken by the animal to run from 0 to 200 cm. We assessed if there was any correlation between the decoder fit and the time the animal took to complete a lap. To do so, we created a histogram of the distribution of time taken to complete a lap in R and UR (Supplementary Fig. 5). For each animal, we divided the laps in UR into those that overlapped with the histogram in R (Matched velocity laps) and those that did not (Slower velocity laps). The average time taken to complete a lap in the matched laps was $7.34 \pm 0.46$ s in R and $7.41 \pm 0.40$ s in UR. The slower speed laps took $19.99 \pm 1.85$ s. Most of the laps belonged to the Matched speed laps and consisted of 70% of the total laps run by all animals in UR. Results are shown in Supplementary Fig. 5.

**Engagement with VR—approach behavior.** In R, mice slowed down as they approached the end of the track. We postulated that if mice were continuing to pay attention to where they were in VR when reward was removed, they would display a similar approach behavior. As a control, we first recorded running behavior of trained animals in the dark ($n = 6$), without any visual cues, to ensure that well trained mice were not displaying a stereotypical behavior independent from VR. To assess approach behavior, instantaneous velocity was calculated at each point along the 2 m track. This velocity trace was then smoothed by averaging it over 5 cm bins. In the dark, there were no signs of stereotyped behavior that looked like approach behavior (Supplementary Fig. 6A). The degree of this approach behavior at the end of the track was calculated as the ratio between lap velocity in the middle (100–150 cm) and end (175–200 cm) of the track as indicated above the traces in each condition. On average (mean ± 95% CI), this ratio was $1.01 \pm 0.03$ in the Dark, $1.3 \pm 0.02$ in the Rewarded condition and $1.22 \pm 0.02$ in the Unrewarded condition. Engaged laps in each animal after licking stops ($RE_{low}$) were then defined as laps where the approach ratio was greater than or equal to mean ± 1.5 * standard deviation of the ratio in the Rewarded Condition (rest were defined as disengaged laps). In total, number of laps in each condition were obtained as follows: $RE_{high}$:Engaged = 90, $RE_{low}$:Engaged = 170, $RE_{low}$:Disengaged = 74. Mean ± 95% CI approach ratio in each condition: $RE_{high}$:Engaged = $1.26 \pm 0.04$, $RE_{low}$:Engaged = $1.3 \pm 0.02$, $RE_{low}$:Disengaged $1.02 \pm 0.03$. To ensure matched behavior in Rewarded and $RE_{low}$ conditions and that the disengaged laps do not skew our results, only engaged laps were extracted from animals for further analysis. Only one animal continued to randomly pre-lick at laps after our definition of lick stop. Of those laps ($n = 10$ laps), 5 were classified as engaged and 5 as disengaged. Reanalyzing the data excluding these laps did not change the decoder error. N = 3/6 animals that went through the R-UR-RR paradigm had enough engaged laps (>12) to define place fields and their results are displayed in Figs. 2 and 3. (see Supplementary Figs. 9 and 11 for all animals).

**Engagement with VR—pupil measures.** To obtain images with dark pupils and high contrast around the borders of the pupils, pupil images were inverted, and their brightness/contrast was adjusted. Pupil area, pupil center of mass (COM), and blinking area were obtained using

FaceMap (v0.2.0, Stringer et al.[80] 2019). Pupil data during blinking periods (frames where blinking area <mean – twice the standard deviation of the blinking area) was removed and the pupil data was interpolated to match the 2-photon imaging frame rate (30 fps). The pupil data was filtered to exclude time periods where the animal was immobile (speed <1 cm/s).

**Pupil area correlation.** To obtain a pupil area trace for each lap, we binned the track into 40 bins (5 cm wide) and calculated the mean pupil area of each bin. For each mouse, the average pupil area of each bin across all laps in the familiar rewarded condition was calculated and served as a template pupil area trace. The pupil area correlation was then measured as the Pearson correlation coefficient between the template pupil area trace and the lap's pupil area trace. High pupil area correlation laps were defined as laps whose pupil area correlation >= mean – 1.5 * std of the pupil area correlation for rewarded laps.

**Mean eye movement.** Eye movement for each frame in a condition was calculated as the difference between the pupil's center position and the mean center position of the pupil during the condition. The mean eye movement for each lap was then calculated.

**Blinking ratio.** Defined as the number of frames defined as blinking periods divided by the total number of frames in each lap.

**Freezing ratio.** Defined as the number of frames where the animal was immobile (speed <1 cm/s) divided by the total number of frames in each lap.

### Defining place fields

Place fields were identified as described in previous studies[44,67,75] with a few key differences. The 2 m track was divided into 40 position bins (each 5 cm wide). The running behavior of the animal was filtered to exclude time periods where the animal was immobile (speed <1 cm/s). Filtering was done to ensure that place cells were defined only during active exploration. In UR, only $RE_{low}$ frames after the licking stopped (see section on "Licking behavior") were included for place cell analysis. Separately, $RE_{low}$:Engaged laps only were included for place cell analysis (Figs. 2 and 3, see "Animal engagement with VR" section). Place fields across the entire track were extracted if they began firing on the track (see clipped cells at the end of the track in Fig. 2b). Cells that began firing at or after reward delivery and during teleportation were excluded from this analysis (although see Reward cells below). Extracted place fields satisfied the following criteria and the same criteria was used for all conditions and all mice: 1. Their width was > 10 cm (except for fields that are clipped at the end of the track). 2. The average ΔF/F was greater than 10% above the baseline. 3. The average ΔF/F within the field was >4 times the mean ΔF/F outside the field. 4. The cell displayed calcium transients in the field on >30% of laps. 5. The rising phase of the mean transient was located on the track. 6. Their p-value from bootstrapping was <0.05[75]. Multiple place fields within the same cell were treated independently.

### Place field parameters

To calculate the various place field parameters, we binned the track into 40 bins (5 cm wide) and measured the mean ΔF/F of each bin. The data of each place field was a Lx40 matrix where L is the number of laps traversed by the animal. For all measures other than out-of-field firing and spatial correlation, transients outside the defined place field region were removed.

**Center of mass (COM).** The COM from all traversals $L$ was calculated as described in ref. 68. Briefly, COM for each traversal was

calculated as,

$$\text{COM}_L = \frac{\sum_i F_i x_i}{\sum_i F_i}$$

where $F$ is the $\Delta F/F$ in each bin $i$ and $x_i$ is the distance of bin $i$ from the start of the track.

**Reliability.** Reliability of a place cell is the consistency with which it fires at the same location across multiple lap traversals. To calculate this, we computed the Pearson correlation between each lap traversal to obtain an $L \times L$ matrix. To obtain the reliability index, the average of this correlation matrix was multiplied by the ratio of number of laps with a significant calcium transient within the field and the total number of laps. The reliability index is 1.0 if the cell fires at the same location in each lap and 0.5 if it fires at the same position but only in half the laps, and so on.

**Out/in place field firing ratio.** This was computed as the ratio between the mean $\Delta F/F$ in bins outside the place field and the mean firing in bins within the place field.

**Width.** Width of the place field was computed as the distance between the spatial bin at which the mean place field rose above 0 and the spatial bin when it decayed back to 0. For place fields at the end of the track that were clipped the end of the place field was considered as the end of the track.

**Firing intensity.** Firing intensity of the place field was calculated as the peak $\Delta F/F$ of the mean place field.

**Population vector correlation.** To determine level of similarity in spatial representations from lap-to-lap in the different conditions, population vector (PV) correlations were calculated. For each of the 40 spatial bins, population vectors were defined as the mean rate of firing for each place cell in that bin. The correlation between the population vector in one lap versus another lap was then calculated and the correlations were averaged over all positions (Fig. 2a).

**K-means clustering.** $K$-means clustering was performed on the calculated lap-wise population vectors. The elbow method was used to determine the optimal number of clusters. For all animals, the method determined this to be 3. $K$-means clustering was performed 1000 times. Each time, the Rewarded cluster was determined as the cluster ID to which most rewarded laps belong to. The probability of all laps (in R, UR and RR) belonging to the Rewarded cluster was then calculated over the iterations (Fig. 2a).

**Spatial correlation with Rewarded condition.** To calculate the consistency of firing of the place cells defined in R across different conditions, we calculated the Pearson correlation coefficient between mean place cell activity defined in R and the mean of the $L \times 40$ matrix of the same cells in other conditions. The within-session correlation was calculated from control animals ($n = 6$ mice). The control rewarded condition (the duration control mice were in this condition matched experimental mice that experienced R-UR-RR) was divided into two halves and the correlation coefficient was calculated between the mean place cell firing in the two halves.

**Place field parameters in DREADD experiments.** All place cells and associated parameters were calculated and quantified as described above for R-UR-RR experiments.

**Reward over-representation**
To compute the density of place cells along the track, the COM of all place fields in all animals were fitted to a gaussian distribution (mean ± standard deviation of the gaussian distribution in cm in different conditions, R: 114 ± 55, UR: 102 ± 54, RR: 112 ± 58, N: 108 ± 53, DREADD Experiments: R: 110 ± 55, UR: 103 ± 54, Before Saline: 113 ± 55, After Saline: 114 ± 54, Before CNO: 113 ± 57, After CNO: 109 ± 56, Before DCZ: 113 ± 56, After DCZ: 106 ± 55) and a uniform distribution to extract regions of place cell overrepresentation (Figs. 3d–f and 4k–m and Supplementary Figs. 11C, 15G, 17E, 18E). To compare changes in place field density across conditions between the middle of the track (50–150 cm) and end of the track (150–200 cm), we divided the middle of the track into 50 cm bins and averaged place cell density across the bins.

**Reward cells**
Reward cells were defined as described in[24]. Briefly, a cell was defined as a reward cell if it fired at the reward zone on the track (40 cm before reward) and around reward delivery (2 seconds before and after reward delivery) in both R and NR. The reward zone on the track was chosen based on the area of high place field density before the reward in R and N (Supplementary Fig. 14A). In total, we found 43 such cells from 6 animals, both on track and around reward delivery (Supplementary Fig. 14A). These cells constituted 0.9% of all active cells recorded. To compare reward cell firing across all conditions, we computed the lap wise firing of these cells in time around reward delivery (Supplementary Fig. 14). Their COM in time around reward delivery, reliability and correlation with R was then calculated similar to place cells.

**Axon imaging analysis**
To characterize the activity of VTA axons, their activity was divided into time and positional bins. For positional bins, the 2 m track was divided into 40 position bins (each 5 cm wide) and the mean fluorescent activity in each bin for every lap was calculated. For time bins, we aligned each lap with the reward delivery and divided the lap into 40 time bins. The average time to reward was 11.9 s ($\pm 0.25$ s, SEM) and the time after reward was 2 s. Therefore, to align each laps reward delivery and maintain roughly equal time bins, the time before reward was divided into 34 time bins and the time after reward was divided into 6 time bins. To account for potential shifts in baseline fluorescence in both position and time binned data, the binned fluorescence data was subtracted by the minimum bin fluorescence for each lap. The binned data was then normalized by dividing by the maximum bin fluorescence for each mouse and pooled across mice. Finally, the average binned fluorescence was calculated for each task condition.

**DA ramp slope and max.** To characterize the ramping activity observed in VTA axons, the maximum and slope of the time binned fluorescence data were calculated for each lap. The maximum was defined as the maximum bin value of the time binned fluorescence data for each lap. The maximum values for each lap were then normalized by dividing by the average maximum value in the Rewarded condition for each mouse. To calculate the slope of the curve in the Rewarded condition, the maximum value near the end of the track (within 15 bins of lap end) and the minimum value near the beginning of the track (within 25 bins of lap start) were determined for each lap. In all other experiment conditions, the range of bins used to find the maximum and minimum values were restricted to the nearest and furthest bins where the maximum and minimum were found in the Rewarded condition for each mouse. A line was then fit to the data points between the defined maximum and minimum values using the matlab fitlm function. The slope of this line was found and normalized by dividing by the average slope in the Rewarded condition for each lap. The slope*max was calculated as the product of the slope and

maximum values for each lap and was normalized by dividing the average slope*max in the Rewarded condition for each mouse. The average maximum, slope and slope*max in each experimental condition were calculated for each mouse.

**Velocity encoding.** To investigate velocity encoding in a VTA axon, we aligned the activity of the axon to motion initiation. Motion epochs were identified as periods where the animal's velocity ≥1 cm/s for at least 1 s. Motion epochs were aligned to motion initiation, or the first frame where velocity ≥1 cm/s. The $\Delta F/F$ data and velocity 2 s prior to motion initiation and 8 s after motion initiation were collected for each motion epoch. Velocity was normalized by dividing by the maximum velocity of each motion epoch. The average $\Delta F/F$ and velocity of all motion epochs was calculated for each experiment condition.

## Statistics

For data distributions, a Shapiro–Wilk test was performed to verify if the data was normally distributed. If normality were true, where applicable, a paired or unpaired Student's $t$ test was used. For non-normal distributions, a paired Wilcoxon signed rank test or an unpaired Mann–Whitney $U$-test was used. To compare between distributions, a two-tailed Kolmogorov–Smirnov (KS) test was used. For samples with five data points or less, only a non-parametric test was used. Multiple comparisons were corrected with Bonferroni post hoc. Throughout the manuscript, boxplots are plotted to display the full distribution of the data. The box in the boxplot ranges from the first quartile (25th percentile) to the third quartile (75th percentile) and the box shows the interquartile range (IQR). The line across the box represents the median (50th percentile). The whiskers extend to 1.5*IQR on either sides of the box and anything above this range is defined as an outlier. Significance tests were performed with and without outliers. P-values calculated without outliers have been displayed in the figure panels. To model the probability distribution in the datasets and get an accurate idea of the data shape, a kernel density estimate was fitted to the data distribution and is shown alongside histograms. Cumulative probability distribution functions were compared using a KS test. We employed estimation statistics to ascertain the level of difference between distributions by using the DABEST (v0.3.1, Data Analysis with Bootstrap-coupled Estimation) package[82]. Estimation plots display the median difference between two conditions against zero difference, with error bars displaying 95% confidence intervals of a bootstrap generated difference (5000 resamples). A kernel density fit (shaded curve) on the resampled difference is also displayed alongside. This difference was compared against zero. Correlations were performed using Pearson's correlation coefficient. Data preprocessing and analysis was done on MATLAB (Mathworks, Version R2018a) and Python 3.7.4 (https://www.python.org/).

## Figure graphics

All figure graphics including Figs. 1a, b, 4b, c, and 5a and Supplementary Figs. 13A and 15A were created using BioRender.com.

## Reporting summary

Further information on research design is available in the Nature Portfolio Reporting Summary linked to this article.

## Data availability

Raw imaging data are extremely large and not feasible for upload to an online repository but are available upon request at sheffield@uchicago.edu. Processed source data for all figures and associated statistical analysis are provided with the paper. Source data are provided with this paper.

## Code availability

Scripts used for data analysis are available on Github (https://github.com/seethakris/HPCrewardpaper).

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

## Acknowledgements
We thank M. Howe, J. Heys, X. Zhuang, J. Yu, and N. Spruston for comments on the manuscript, D. Freedman for comments on the Cover Letter, M. Rosen for assistance on data analysis, H. Macomber for behavior data of mice running in the dark and members of Sheffield lab for manuscript comments and useful discussions. This work was supported by The Whitehall Foundation, The Searle Scholars Program, The Sloan Foundation, The University of Chicago Institute for Neuroscience start-up funds and the NIH (1DP2NS111657-01) awarded to M.S. and a T32 training grant (T32DA043469) from National Institute on Drug Abuse awarded to C.H. and S.K.

## Author contributions
S.K. and M.S. conceived and designed the experiments. C.H. and C.C. performed surgeries. S.K., C.H., and C.C. collected the data. S.K. wrote the analysis code and analyzed hippocampal data. C.H. collected and analyzed VTA axon data. S.K., C.H., and M.E.J.S. interpreted the data and wrote the manuscript.

## Competing interests
The authors declare no competing interests.
