## [Peer Review File · Nature Communications]

Reward expectation extinction restructures and degrades CA1 spatial maps through loss of a dopaminergic reward proximity signalEditorial Note: This manuscript has been previously reviewed at another journal that is not operating a transparent peer review scheme. This document only contains reviewer comments and rebuttal letters for versions considered at *Nature Communications*.

REVIEWERS' COMMENTS

Reviewer #1 (Remarks to the Author):

The response to reviews is comprehensive and the authors have addressed all of my concerns. This paper meets the standards for publication in Nature Communications.

Reviewer #2 (Remarks to the Author):

The authors have done a good in addressing the previous concerns.

I have a couple of relatively minor comments.

Extended Data 15 should be modified to make clear what the colour codes for the experimental groups correspond to. The figure legends should also state the n value and test-statistic as well as the p values for the reported statistical tests.

For Figure 4f and Extended Data 15d it should be made clear if the t-tests are applied at the level of data from individual neurons, or by comparing per animal averages. If the former then some justification for neglecting dependencies between neurons from the same animal should be given, or either a nested test or an animal-level test used instead.

Reviewer #3 (Remarks to the Author):

The authors have addressed all my previous comments.

Regarding the overrepresentation of reward locations and my initial comment#2 that it might arise from non-spatial spiking activity associated with resting state because of a too low speed threshold, it is pertinent that the new analysis of Extended Data Fig.14A is supporting the spatial nature of reward firing fields as many reward-encoding cells encode a different position in the other context. The authors might mention in the text that the reward place fields are largely context-specific, like typical place fields, to strengthen their point.

Reviewer #4 (Remarks to the Author):

I would like to thank the authors for the thorough revision of the manuscript.

The revision addresses most of my concerns. In their overall improved manuscript, the authors have also now clearly delineated the impact of their study in the context of existing literature, which was a major concern shared across reviewers (Reviewer 1).

One of the few remaining points is my suggestion of projection specific perturbations (Major 7). The authors agree that projection specific VTA perturbations would support much better their imaging data. However, given the technical difficulties of this experiment cited by the authors, I understand that obtaining such data will go beyond the scope of this revision.

Thus, in summary, am I looking forward to seeing this paper published.

Response to second round of Peer-Review on manuscript NCOMMS-22-20171-T

Reviewer #1 (Remarks to the Author):

The response to reviews is comprehensive and the authors have addressed all of my concerns. This paper meets the standards for publication in Nature Communications.

We thank the reviewer.

Reviewer #2 (Remarks to the Author):

The authors have done a good in addressing the previous concerns.

We thank the reviewer.

I have a couple of relatively minor comments.

Extended Data 15 should be modified to make clear what the colour codes for the experimental groups correspond to. The figure legends should also state the n value and test-statistic as well as the p values for the reported statistical tests.

We have updated the figure and legend to address these concerns.

For Figure 4f and Extended Data 15d it should be made clear if the t-tests are applied at the level of data from individual neurons, or by comparing per animal averages. If the former then some justification for neglecting dependencies between neurons from the same animal should be given, or either a nested test or an animal-level test used instead.

It is true that we applied the test to the level of data from individual neurons and we thank the reviewer for noting the nested dependencies in such a case. Therefore, we have now instead used a KS-Test as our main intention is to determine whether the distributions are similar.

Reviewer #3 (Remarks to the Author):

The authors have addressed all my previous comments.

Regarding the overrepresentation of reward locations and my initial comment#2 that it might arise from non-spatial spiking activity associated with resting state because of a too low speed threshold, it is pertinent that the new analysis of Extended Data Fig.14A is supporting the spatial nature of reward firing fields as many reward-encoding cells encode a different position in the other context. The authors might mention in the text that the

reward place fields are largely context-specific, like typical place fields, to strengthen their point.

We thank the reviewer for this comment and have added a sentence to reflect this point to lines 305-306:

“Thus, place fields near reward sites act like typical place fields, i.e. they are largely context-specific.”

Reviewer #4 (Remarks to the Author):

I would like to thank the authors for the thorough revision of the manuscript.

The revision addresses most of my concerns. In their overall improved manuscript, the authors have also now clearly delineated the impact of their study in the context of existing literature, which was a major concern shared across reviewers (Reviewer 1).

One of the few remaining points is my suggestion of projection specific perturbations (Major 7). The authors agree that projection specific VTA perturbations would support much better their imaging data. However, given the technical difficulties of this experiment cited by the authors, I understand that obtaining such data will go beyond the scope of this revision.

Thus, in summary, am I looking forward to seeing this paper published.

We thank the reviewer for all their comments that helped improve the paper.